# The unicellular NUM v.0.91: A trait-based plankton model evaluated in two contrasting biogeographic provinces.

Trine F. Hansen[1*], Donald E. Canfield[1,2,3], Ken H. Andersen[4], Christian J. Bjerrum[5]

[1]Nordcee, Department of Biology, University of Southern Denmark, Odense M, Denmark
[2]Danish Institute of Advanced Studies (DIAS)
[3]Petrochina, Beijing, China
[4]Center for Ocean Life, National Institute of Aquatic Resources, Technical University of Denmark, Kongens Lyngby, Denmark
[5]Department of Geosciences and Natural Resource Management, University of Copenhagen, Copenhagen, Denmark
*Now at Center for Ocean Life, National Institute of Aquatic Resources, Technical University of Denmark, Kongens Lyngby, Denmark

*Correspondence to*: Trine F. Hansen (trfri@aqua.dtu.dk)

**Abstract**

Trait-based models founded on biophysical principles are becoming popular in planktonic ecological modeling, and justifiably so. They allow for slim, efficient models with a significant reduction in parameters, well suited for modeling the past and future climate changes. In their idealized form, trait-based models describe the ecosystem in one set of parameters defined by first principles, rooted in physics, chemistry, geometry, and evolution. The result is an emerging ecosystem 20 defined by physical and chemical limitations at the cell level. At present, however, a significant part of these parameters is not fully constrained, which potentially introduces considerable uncertainty to the model results. Here, we investigate how these parameters influence the ecosystem structure of one of the simplest trait-based models, the Nutrient-Unicellular-Multicellular (NUM) model. We describe the unicellular module of the NUM model and through an extensive parameter sensitivity analysis, we demonstrate that the model - with a large span in parameters – can capture the general features of the 25 pico-, nano-, and micro planktonic ecosystem in a high-productivity upwelling system. We demonstrate that it is possible to narrow the range of parameters to get a stable, acceptable, solution. Finally, the model responds correctly in an oligotrophic downwelling system using parameters fitted to the upwelling system. Our analysis demonstrates that the unicellular module of the NUM model is broadly accessible without detailed knowledge of the parameter settings, and that the first-principal approach is well suited for modeling poorly resolved region and ecosystem evolution during current and deep time climate 30 change.

# 1 Introduction

Trait-based models are becoming an important tool for understanding the spatial and temporal pattern in planktonic ecosystem structure (e.g. Follows et al., 2007; Dutkiewicz et al., 2021; Ward et al., 2019; Eckford-Soper et al., 2022). Rooted in first principles of biophysics and biochemistry, trait-based models alleviate many of the caveats that confine traditional functional planktonic ecosystem models: they allow for large-scale ocean domains without the need for adding increased complexity; they reduce the amount of parameter tuning; and they allow for modeling evolution in the past and future, under climate change where ecosystems are different from the ones we know today (Reinhard et al., 2020; Sauterey et al., 2017; Wilson et al., 2018; Archibald et al., 2022).

There are a variety of approaches to trait-based modeling. For most of the trait-based planktonic ecosystem models, size is used as a master trait, as it scales with many of the cells processes and rates (Ward et al., 2019; Sauterey et al., 2015; Andersen et al., 2015). One particularly simple size-based plankton model is the Nutrient-Unicellular-Multicellular (NUM) model (Andersen and Visser, 2023; Serra-Pompei et al., 2020; Serra-Pompei et al., 2022). The NUM model is founded in the biophysical and chemical processes of the cell, scaled up to community level (Fig. 1). With the cell processes at the center, the result is a simple and fast model where size-spectrum and rates of photosynthesis, as well as uptakes of nutrients, dissolved organic carbon (DOC) and food (phagotrophy), emerges from the specific physical conditions of the oceanographic conditions (Andersen and Visser, 2023; Serra-Pompei et al., 2020).

Despite the simplicity of the NUM model, it - like any other model – relies on a set of parameters (Table 1). In principle, these parameters are universal and common for all organisms; however, they are not all well established. Some parameters are well defined by cell physiology, e.g., the maximum diffusive nutrient affinity coefficient ($\alpha_D$) that is limited by cell surface area, but many have a range of uncertainty that emerges from natural cell variability or from a limited understanding of the parameter. As with any model, the output of the NUM model reflects the parameter choices. It is still, however, unclear how much the parameters influence the result, how much tuning the model require and how well the model transfers between sites with the current parameter uncertainty.

In this article we describe the unicellular module of the NUM model and evaluate the model's ability to capture well-established key ecosystem descriptors, its robustness, geographical transferability, and the relative importance of the underlying parameters. Specifically, we start by evaluating the model's ability to capture the size structure of the planktonic biomass at the California Current Ecosystem (CCE) (California-Current-Ecosystem-Lter and Landry, 2019; Taylor and Landry, 2018), using default model parameters. Hereafter we evaluate how the parameter uncertainty effects the model sensitivity. We conclude by applying the identified optimal parameter values for the CCE in a test of the model's geographical transferability to the ALOHA station north of Oahu, Hawaii (Pasulka et al., 2013; Taylor and Landry, 2018).

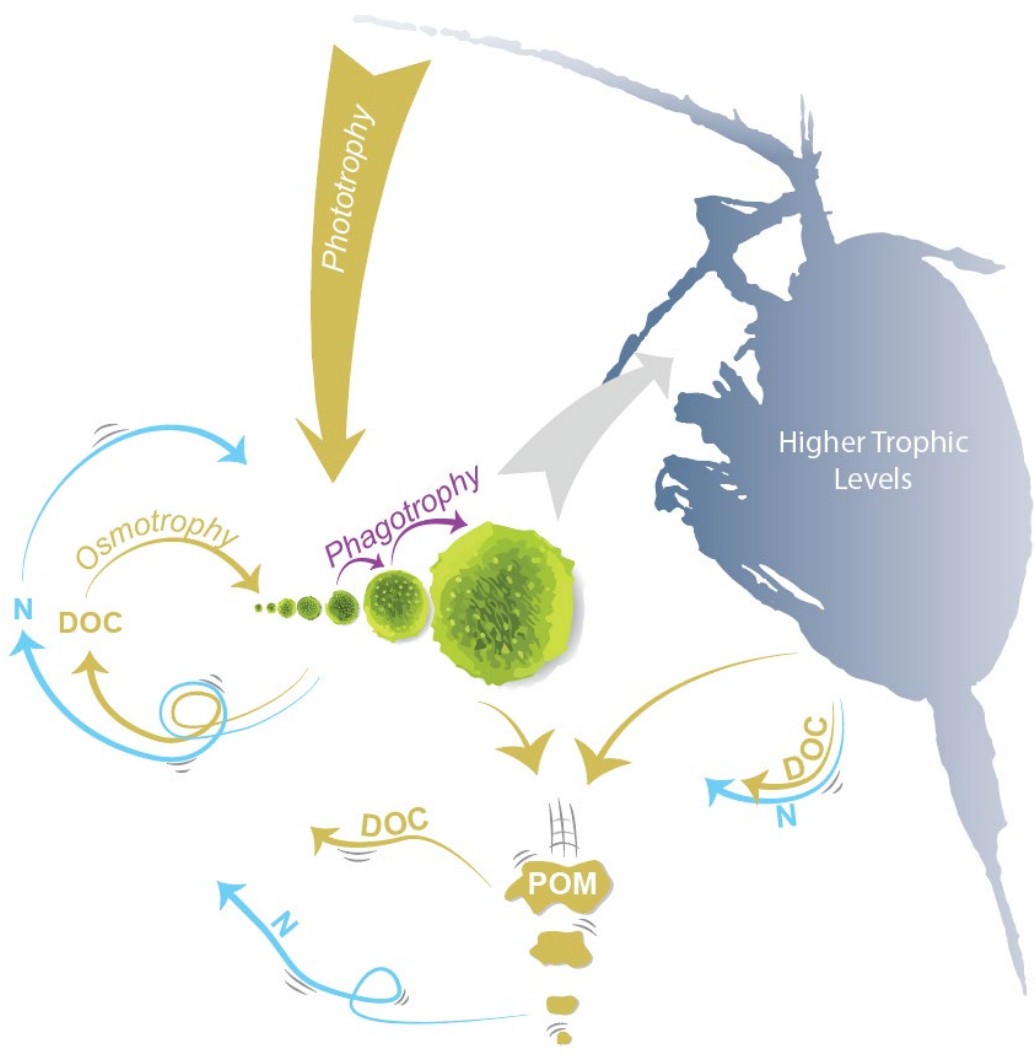

**Figure 1: Schematic of the unicellular module of the Nutrient-Unicellular-Multicellular (NUM) model. The unicellular organisms are here represented by 7 size-classes of organisms that can all get their nutrients and carbon from osmo-, photo- and phagotrophy. The uptake rates depend on the biophysics of the cell and the environmental availability of dissolved organic carbon (DOC), nutrient (N), light and food (smaller cells). Higher trophic levels are here parameterized as feeding on a specific size range of cells. Exudation, viral lysis, assimilation losses, and higher trophic level losses replenish the nutrients and carbon. Losses from sloppy feeding by phagotrophy and higher trophic level are re-introduced as particulate organic matter (POM) that sinks down through the water column and is remineralized into DOC and N. The model formulations are listed in Supplement S1.**

**Table 1: Parameters used in this study. Reference values are based on arguments from Andersen and Visser (2023) and standard values used in the NUM model setup.**

| Parameter | | Unit | reference value | Parameter range min | Parameter range Max | Parameter confidence[4] |
|---|---|---|---|---|---|---|
| carbon density | $\rho$ | µgC µm$^{-3}$ | 0.4×10$^{-6}$ | 0.3×10$^{-6}$ | 0.5×10$^{-6}$ | 1 |
| C:N mass ratio | $\rho_{C:N}$ | gC gN$^{-1}$ | 5.68 | 2.7 | 8.7 | 1 |
| **Cell rate parameters** | | | | | | |
| Diffusive affinity coefficient | $\alpha_D$ | l µm$^2$ d$^{-1}$ (µgC)$^{-1}$ | 0.972 | 0.75 | 1.3 | 1 |
| Diffusive affinity crossover | $r^*_D$ | µm | 0.4 | 0.1 | 5 | 1 |
| Light affinity coefficient | $\alpha_L$ | (d µmol m$^{-2}$ s$^{-1}$)$^{-1}$ µm | 0.3 | 0.05 | 1.5 | 2 |
| Light affinity crossover | $r^*_L$ | µm | 7.5 | 2.5 | 20 | 2 |
| Light uptake efficiency | $\varepsilon_L$ | unitless | 0.8 | 0.1 | 0.9 | 2 |
| Clearance rate | aF | l d$^{-1}$ µgC$^{-1}$ | 1.8×10$^{-2}$ | 8.23×10$^{-4}$ | 0.4455 | 2 |
| Max. phagotrophic coefficient | cF | µm d$^{-1}$ | 30 | 10 | 50 | 3 |
| Assimilation efficiency | $\varepsilon_F$ | unitless | 0.8 | 0.1 | 0.9 | 1 |
| Passive losses coefficient | $c_{passive}$ | unitless | 0.03 | 0.01 | 0.1 | 2 |
| Maximum synthesis coef. | $\alpha_{max}$ | d$^{-1}$ | 1.5 | 0.1 | 2.1 | 2 |
| Basal metabolism coef. | $\alpha_R$ | unitless | 0.1 | 0.045 | 0.22 | 2 |
| **Prey encounter** | | | | | | |
| predator-prey mass ratio | $\beta$ | unitless | 500 | 300 | 700 | 2 |
| predator-prey width | $\sigma$ | unitless | 1.3 | 0.9 | 1.7 | 2 |
| **community model parameter** | | | | | | |
| DOC remin. of feeding | $\gamma_F$ | unitless | 0.1 | 0.1 | 0.9 | 3 |
| DOC remin. of viral lysis | $\gamma_2$ | unitless | 0.5 | 0.1 | 0.9 | 3 |
| Viral lysis mort. coefficient | $\mu_{v0}$ | unitless | 4.0×10$^{-3}$ | 4.0×10$^{-4}$ | 4.0×10$^{-2}$ | 3 |
| Size of HTL mortality[1] | $m_{HTL}$ | µgC | 0.1 | 0.001 | 0.1 | 2 |
| HTL mortality coefficient | $\mu_{htl}$ | d$^{-1}$ | 0.1 | 1.0×10$^{-2}$ | 0.25 | 3 |
| **Particulate organic matter (POM)** | | | | | | |
| POM sinking coefficient[3] | $v_1$ | m d$^{-1}$ | 100 | 1 | 200 | 2 |
| Inverse solubilization length scale[2] | a | m$^{-1}$ | 0.004 | 0.002 | 0.006 | 2 |
| Fraction of HTL mort. to POM | $\gamma_{HTL}$ | unitless | 0.5 | 0.1 | 0.9 | 2 |
| **Fixed parameters** | | | | | | |
| Membrane thickness | $\delta$ | nm | 50 | | | 2 |
| Light attenuation by water | $k_w$ | m$^{-1}$ | 0.05 | | | 2 |
| Light attenuation by POM[3] | $k_{POM}$ | m$^2$ mg C$^{-1}$ | 3×10$^{-5}$ | | | 2 |
| POM sinking exponent[3] | $v_2$ | md$^{-1}$ | 0.2 | | | 2 |

[1]*The size of HTL mortality is between 100 and 10000 times smaller than the largest cell size.*

[2]*Fennel et al. (2001)*

[3]*POM was not included in previous versions of the NUM model and the parameters written in the reference value signify the values used in the initial evaluation of the model. Based on arguments in supplement S1, a $k_{POM}$ value of 3×10$^{-5}$ m$^2$ mg C$^{-1}$ is*

*used for all simulations in this article. The choice in POM sinking coefficient and exponent result in a sinking speed of 0.01-3 m day$^{-1}$ for the smallest POM size classes and 1-200 m day$^{-1}$ for the largest, using the formulation for POM sinking in supplement S1.*

[4]*Qualitative parameter uncertainty ranging between 1 (low) and 3 (high) cf. Andersen and Visser (2023). Parameter uncertainty stem from limited understanding of processes and/or empirical evidence.*

## 2 Model description

### 2.1 The Nutrient-Unicellular-Multicellular model framework

The NUM model is built on an additive model framework that relies on formulations of the fundamental properties of the organism (Andersen and Visser, 2023; Serra-Pompei et al., 2020; Serra-Pompei et al., 2022). The NUM model initially included copepods and protists as the unicellular and multicellular components of the model, along with nutrient (N) and fecal pellets (Serra-Pompei et al., 2020). Serra-Pompei et al. (2020) implemented the model in MATLAB with a chemostat setup. Later, Serra-Pompei et al. (2022) coupled the NUM model to a transport matrix, enabling both water column and global simulations. A major update of the NUM model resulted in the current version where the core NUM model was translated from MATLAB to FORTRAN95. The model can be run directly from FORTRAN but can also be initialized from MATLAB and from R, which opens the model to users without FORTRAN experience. In this update, the NUM framework was extended to include a DOC module and a particulate organic carbon (POM) module.

The NUM model can be used in three different setups; It can be used in a global simulation where the NUM model is coupled to a transport matrix that provide advection, diffusion, and temperature for the simulation (Khatiwala, 2007); It can be used in a chemostat setup with a constant mixing rate and deep nutrient concentration; and finally, as we do here, it can be used in a water column simulation where temperature and diffusion at single location is extracted from the transport matrix. Here, we describe and evaluate the unicellular organisms and the particulate matter, and we will therefore limit the description of the NUM framework to these parts. The model formulations are provided in Supplement S1. Section 2.2 describes the unicellular module and parameters while section 2.3 describes the new simple DOC and POM modules and the associated parameters.

### 2.2 The unicellular module

The backbone of the NUM is the unicellular module that comprises the classic functional groups of phytoplankton, osmotrophic bacteria, and zooplankton. While unicellular organisms span many orders of magnitude in size, across all types of trophic strategies (feeding mechanisms), they are all described with one set of parameters in the unicellular subroutine of

NUM. Here, the cell may be visualized as one type of organism – we refer to as a *generalist* - that is essentially a mixotroph in the sense that it is able to perform osmotrophy (diffusive uptake of DOC), photosynthesis and phagotrophy (food consumption) to gain nutrient and carbon. The generalist can utilize all three trophic strategies at the same time. However, the yield from each of these strategies depends on the size of the generalist and on the surrounding environmental conditions (light level, nutrient and dissolved organic carbon concentration, food, etc.). The model contains several of these generalists with the only difference being the size of the organism, defined in logarithmic size-bins of mass, *m*. The output of the generalist subroutine is the biomass of each of the generalist size bins and the associated rates of phototrophy ($J_{\text{auto}}$), osmotrophy ($J_{\text{osmo}}$) and phagotrophy ($J_{\text{phag}}$). This approach makes the unicellular module especially well adapted to handle mixotrophic organisms. In the following subsections, we will go through the most important processes controlling the generalist growth and size structure. The aim of this section is to give the reader an understanding of the mechanisms that control the organism and a sense of the parameters that are evaluated in this study. The important parameters are highlighted in bold in the text below. The following sections summarize the more detailed description of the model given in Serra-Pompei et al. (2020) and Andersen and Visser (2023).

### 2.2.1 Resource uptake

The organism's affinity for (*meaning its ability to take up*) dissolved organic matter and nutrients ($a_{\text{D}}$), light ($a_{\text{L}}$) and for food phagotrophy ($a_{\text{F}}$) is dependent on its size. The affinities for uptake of these resources are determined by the encounter rate (*how much resource the generalist is in contact with*) and the assimilation rate (*how fast it can take up the resource it encounters*).

The affinity for diffusive uptake of nutrients and dissolved organic carbon (DOC) is modeled as a crossover between two size regimes: large and small organism size. For large organisms, the limiting factor is the rate of diffusion towards the outer cell membrane. In contrast for smaller organisms, it is the numbers of cell porter channels that transport the resource across the cell membrane (Eq. (1), all equations referred to are listed in Table 2). The parameter $r^{*}_{D}$ determines the organism size where the crossover between the two regimes occurs, and the diffusive affinity coefficient, $\boldsymbol{\alpha_{D}}$, defines the upper limit of the diffusive encounter.

The affinity for uptake of carbon through photosynthesis, $a_{\text{L}}$, is also modeled as a crossover between two regimes (Eq. (2)). For small organisms $a_{\text{L}}$ is dependent on the organism's mass, while for larger organisms, where light harvesting complexes create internal shading, $a_{\text{L}}$ is dependent on the cell surface area. The parameter $r^{*}_{L}$ determines the crossover size between the two regimes. The parameter $\boldsymbol{\alpha_{L}}$ is defined as $\alpha_{L} = 3y/(4\rho)$, where $y$ is the quantum-yield (describing the efficiency of the process relative to the available photons) and $\rho$ is the carbon density of the individual cell (cf. Andersen and Visser, 2023). The light uptake efficiency ($\boldsymbol{\varepsilon_{L}}$) is a fraction that defines how efficient the organism is at utilizing the available light.

Phagotrophy is modelled as a hyperbolic curve where an increase in the amount of prey ingested increase with the prey density, until saturation of prey ingestion occurs. Such ecological type-II functional response has a constant affinity (the

clearance rate, $a_F$) and a maximum assimilated phagotrophic uptake that is dependent on the assimilation efficiency ($\varepsilon_F$) and the maximum phagotrophic coefficient ($c_F$) (Eq. (3)).

### 2.2.2 Synthesis, respiration, and losses

    The generalist might be able to take up more nutrient and carbon than it is able to synthesize. The rate of biosynthesis is controlled by the maximum synthesis coefficient ($\alpha_{max}$, Eq. (4)). Nutrients and carbon in excess leaks out of the cell. Beside

the resource uptake, the organism passively leaks carbon and nutrients through the cell membrane. This process is modeled as a constant, $c_{passive}$, divided by the radius of the organism (Eq. (5)). Finally, the organism's respiration rate is modeled as a fraction of the maximum synthesis coefficient (Eq. (6)). This is called the basal metabolism coefficient, $\alpha_R$.

### 2.2.3 Prey-predator interactions

    The generalist is potential prey for two groups: other larger generalists and predators from higher trophic levels. The

generalist's internal prey-predator relationship is based on the two parameters, $\beta$ and $\sigma$ (Eq. (7)). $B$ defines the mean mass ratio between the prey and the predator. The parameter $\sigma$ defines the wideness of the preferred size range that a predator prey on. The mortality from higher trophic levels is likewise defined by two parameters: $m_{HTL}$, that defines the lower limit (expressed as mass) of organisms that are preyed upon by higher trophic levels, and the higher trophic level mortality coefficient, $\mu_{HTL}$, that defines the rate of predation by higher trophic levels. Lastly, the generalists undergo viral lysis. The

rate is controlled by the parameter $\mu_{v0}$ and dependent on the logarithmic size bin length (Eq. (8))

### 2.3 Dissolved organic carbon and particulate organic matter

    This version of the NUM incorporates both dissolved and particulate matter in a simplified approach (Fig. 1). Dissolved nutrients, both inorganic and organic N containing, are modelled as one dissolved N pool, while dissolved organic carbon

(DOC) is modelled separately. The particulate matter (POM) contains both C and N in a fixed ratio. Dead cells, feeding losses, and higher trophic level mortality produce both particulate organic matter (POM) and dissolved constituents (DOC and N). Note, that the choice of pooling inorganic and organic N in a single pool means that the microbial consumption/remineralization of N is not explicitly resolved as dependent on osmotrophy. In contrast, consumption of DOC as an energy source for heterotrophic osmotrophy is explicitly modelled as presented above (section 2.2.1). The pool of DOC

in this model represents "labile" DOC. The division between the particular and dissolved fractions are determined by the $\gamma$ parameters ($\gamma_2$, $\gamma_F$ and $\gamma_{HTL}$), which describe how much of each flux (mortality, feeding losses, and higher trophic level mortality) are routed to the dissolved fractions, with the remaining losses transferred to POM. Particulate organic matter is here divided into two different size fractions (a number that can readily be increased in future applications). POM derived from dead cells and feeding losses is transferred to the largest POM size fraction, which is smaller than the size of the

original cell. POM from higher trophic level mortality is transferred into the largest POM size fraction. POM sinks with a

size-dependent velocity, described as a power function with the parameters $v_1$ and exponent $v_2$ (Eq. (9)). POM is assumed to remineralize directly to the dissolved N and DOC pools. This process of remineralization is not explicitly microbial cell related in the model but occurs at a constant rate determined by the inverse of the solubilization length scale (**a**) as $rem_{POM} = aw_{POM}$. The model formulation of nutrient, along with DOC and POM modules are given in Supplement S1.


**Table 2: Equations used for the unicellular submodule. Full model description is given in Supplement S1.**

| | | |
|---|---|---|
| Affinity for nutrients and dissolved organic matter. | $a_D = \alpha_D r^{-2} \dfrac{1}{1 + (\frac{r}{r_D^*})^{-2}} m$ | Eq.1 |
| Affinity for carbon uptake through photons | $a_L = \dfrac{\alpha_L}{r}(1 - e^{-\frac{r}{r_L^*}})(1 - v)m$ | Eq.2 |
| Rate of phagotrophy | $J_F = \epsilon_F c_F r^{-1} \dfrac{a_F F}{a_F F + c_F / r} m$ | Eq.3 |
| Maximum biosynthesis rate | $J_{max} = \alpha_{max}(1 - v)m$ | Eq.4 |
| Passive losses | $j_{passive} = c_{passive} r^{-1}$ | Eq.5 |
| Respiration rate | $J_R = \alpha_R \alpha_{max} m$ | Eq.6 |
| Size preference for predation | $\varphi = \exp\left[-\dfrac{ln^2(\frac{m}{\beta m_{prey}})}{2\sigma^2}\right]$ | Eq.7 |
| Viral lysis[1] | $\mu_v = \dfrac{\mu_{v0}}{\log(\frac{m^+}{m^-})}$ | Eq.8 |
| Sinking of particulate organic matter | $w_{POM} = v_1 m^{v2}$ | Eq.9 |

*[1]$m^+$ and $m^-$ is mass of the upper and lower limit of the size bin*

## 3 Modelling approach

In this article, we are using the water-column setup of the NUM model to simulate the conditions at the southern California Current Ecosystem (CCE) and Station ALOHA. We initially perform a general validation of the model with default parameters against the mean biomass size spectrum and nutrient profile for the two locations. The subsequent analysis is aimed at understanding the model's performance, robustness, transferability, and parameter sensitivity.

The investigation has two levels: an overall broad random parameter evaluation followed by three more detailed statistical sub-analyses. The first-level parameter study is comprised of 100,000 simulations with quasi-random input parameters in the range defined in Table 1. Of the 23 free parameters, several are assigned a span of several orders of magnitude, which is computationally demanding but enables a genuine investigation of the solution space and variability for the model. We use Latin hypercube sampling scheme for all 23 parameters to ensure an even spread in the entire parameter space (Mckay et al., 1979; Stein, 1987) and evaluate the model performance by comparing the results with observations, using a set of statistical matrices that will be described below. We moreover use this first-level parameter study and evaluation to identify optimized parameter combinations that result in good model fit to CCE observations. These optimized parameter combinations define a *restricted parameter spans* that permit us to make three additional statistical subanalysis for CCE. The first subanalysis is a set of 10,000 simulations where input parameters are quasi-randomly sampled with the Latin hypercube sampling scheme within the *restricted parameter spans*. This subanalysis allow us to determine if only very specific combination of parameter results in good model fit or if model performance is increased by simply reducing the parameter span. The second subanalysis is a set of local sensitivity analyses where the model's sensitivity toward each of the parameters is evaluated separately with outset in an initial parameter combination (Zhou and Lin, 2008). The local sensitivity analysis is made with outset in the initial parameter combinations that performs best for CCE. Each of the parameters are successively varied in 50 evenly distributed intervals within the *restricted parameter span*. This subanalysis showed that several of the parameters results in systems bifurcation points where the model solution changes abruptly. While being extremely interesting, the detailed analysis of such bifurcation points is beyond the current scope and remains a prospect for future analyses. The subanalysis also showed that most parameters are highly coupled in term of ecosystem sensitivity, where the effect of individual parameters are intertwined and result in a highly non-linier system. The sensitivity analysis with a specific parameter outset yielded nearly equally sensitive to almost all parameters whereas with a different parameter outset, $\varepsilon_F$ was the absolute most important parameter. Because of these highly non-linier parameter interactions, local sensitivity studies give little added information about the model performance. We have added two of these seven tests in Supplement S4. The third subanalysis is a global variance-based sensitivity analysis using Sobol's method and sensitivity index (Bilal, 2014; Sobol, 1993, 2001). The global variance-based sensitivity analysis not only accounts for the effect of each individual parameter on the modeled result (the first-order effect) but also more interestingly, the effects of the parameter through its interactions with other parameters (total effect) (Bilal, 2014; Zhou and Lin, 2008). The global sensitivity analysis is made following Bilal (2014), as a set of 20,000 simulations with parameter combinations based on random sampling of the *restricted parameter spans*. Then, for each of the 20,000 simulations we step through the parameters and perform two simulations: (1) the parameter in question is kept at its value while the other 22 parameter are selected quasi-randomly within the restricted parameter span, and (2) the parameter in question is randomly selected in the parameter span while all other parameters are kept at their values (Bilal, 2014; Sobol, 2001). A step-by-step description for the process for setting up the global sensitivity analysis is included in Supplement S5.

The model evaluation and statistical test against the CEE permit us to identify seven optimized parameter combinations that result in a good model fit to observations for the CCE. We then finally evaluate how the model performs within the *restricted parameter spans* at Station ALOHA that, with is different physical and chemical conditions, represent an oligotrophic downwelling system. These results are evaluated against a first-level parameter study at ALOHA with 100,000 quasi-random parameter combinations.

## 3.1 Observational data

Compilations of the composition of phytoplanktonic communities have illuminated some systematic trend in the size distribution of planktonic organisms as a function of chlorophyll and autotrophic biomass concentration ($AC_{bio}$) (Taylor and Landry, 2018; Maranon et al., 2012; Ward et al., 2014). Analyses across various provinces in the Atlantic and in the North Pacific broadly reveal that, when chlorophyll a (Chl-*a*) or primary production is low, ~40% of the biomass is dominated by picophytoplankton (0.2–2 µm), irrespective of temperature. As Chl-*a* increases, microphytoplankton (>20 µm) increase in biomass and dominate when Chl-*a* is high. Nanophytoplankton (2–20 µm) is intermediate between pico- and microphytoplankton at both low and high Chl-*a*. Similar trends are present at sub-regional or local scale in detailed work that is described below (Taylor and Landry, 2018; Taylor et al., 2015; Goericke, 2011) (Fig. 2). Because of the apparent pervasiveness of these trends and characteristic of the planktonic community in marine ecosystems, size structure represents an excellent test for the model's adaptability across oceanographic regimes.

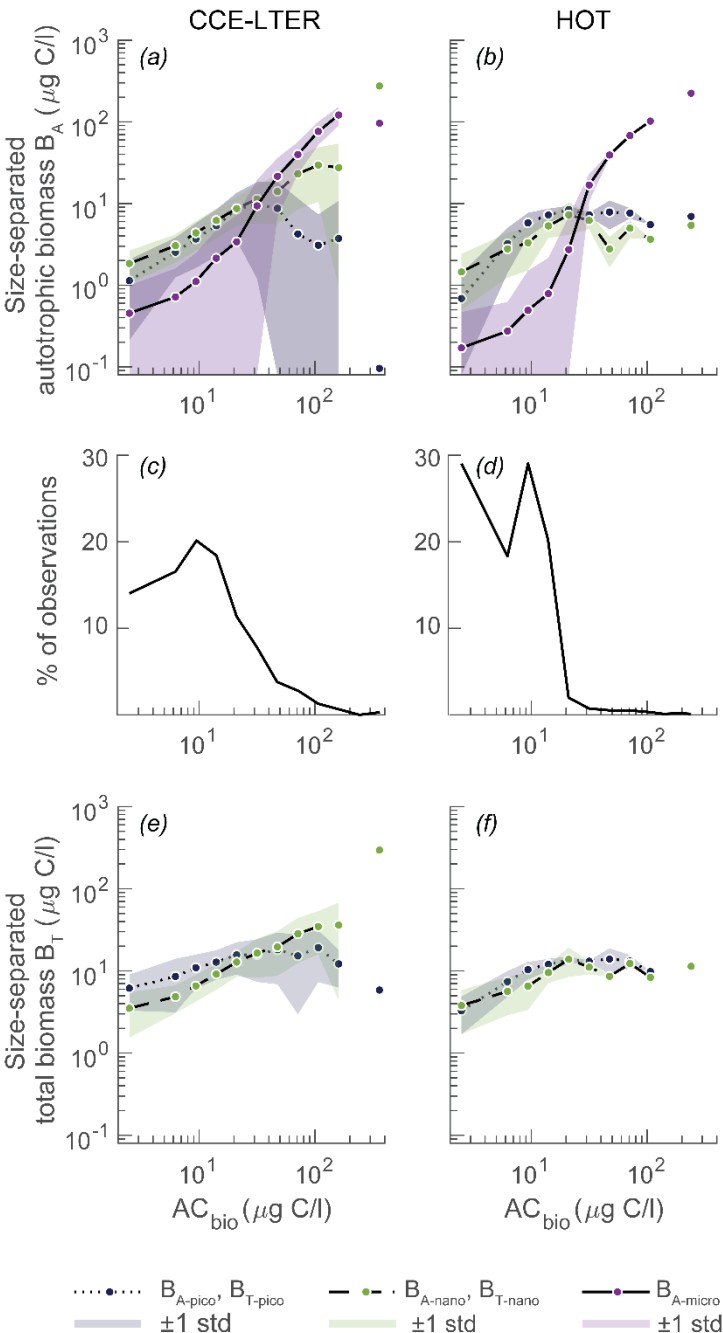

**Figure 2: Observed plankton biomass as function total autotrophic biomass (AC$_{bio}$). (a, b) Mean biomass of pico- (< 2 μm, B$_{A-pico}$), nano- (2-20 μm, B$_{A-nano}$) and microautotrophs (> 20 μm, B$_{A-micro}$) at (a) California current ecosystem (CCE) with upwelling, (b) Hawaii ocean time series (HOT) with downwelling conditions. Data are compiled from 0-200 m depth from 2004 to 2011 and has been binned in logarithmically distributed bins. (c, d) Number of observations per bin at respective sites. (e, f) Total picoplankton (B$_{T-pico}$) and nanoplankton (B$_{T-nano}$) biomass, respectively at CCE and HOT, is the sum of autotrophic and heterotrophic biomass.**

**Data and binning method from Taylor and Landry (2018) and references in text. Note that CEE relative to HOT is more eutrophic reflected by more data in higher $AC_{bio}$ bins (c vs. d).**

Here, we compare the model result to size-spectrum data gathered from the southern California Current Ecosystem (CCE) as a part of the California Current Ecosystem Long Term Ecological Research (CCE-LTER) and from Station ALOHA, the long-term Hawaii Ocean Time series (HOT) (Taylor and Landry, 2018; Pasulka et al., 2013; California-Current-Ecosystem-

Lter and Landry, 2019) (Fig. 2). These two sites reflect distinctly different oceanographic regimes: coastal upwelling with eutrophic conditions at CCE and downwelling oligotrophic open-ocean waters at Station ALOHA. Both sites have been sampled for epifluorescence microscopy and flow cytometry regularly in the years 2004 to 2011, resulting in large datasets of biomass abundance, size structure and planktonic composition. The phytoplanktonic size structure of the two sites show many of the same features as the large-scale compilations of planktonic size distribution: Pico- and nano- autotrophic

organisms dominate the size spectrum at low autotrophic carbon biomass ($AC_{bio}$) where the concentrations of microautotrophic organisms are very low (Fig. 2a, b) (Taylor and Landry, 2018; Maranon et al., 2012; Ward et al., 2014). The concentration of all three size classes increases with increasing $AC_{bio}$, however, the autotrophic microplankton concentration increases faster than the smaller size groups and become dominant at intermediate levels of $AC_{bio}$ (approximately 20 $\mu gCl^{-1}$). Microautotrophic plankton continue to increase in a power law fashion for both observational

datasets. In contrast, the pico- and nanoautotrophs increase as a function of $AC_{bio}$ is different at the two sites. The CCE-LTER dataset follows the global tendency of a continued increase in nano-autotrophs while the pico-autotrophs decrease toward high $AC_{bio}$. The HOT observations show a steadier concentration for both pico- and nanoautotrophs across $AC_{bio}$ concentrations, but with a small decrease in nanoautotrophs at high $AC_{bio}$. While the two sites show many of the same features we note that high autotrophic biomass concentrations are much more frequently observed at CCE than at Station

ALOHA (Fig. 2, c-d). However, it is only in approximately two percent of the observations from the CCE-LTER dataset that autotrophic biomass has been measured as high as 100 µgC/l. At Station ALOHA, only four percent of the observations has autotrophic biomass concentrations of 30 µgC/l.

As explained above, the unicellular subroutine of the NUM framework calculates the rate of nutrient and carbon uptake $J_{auto}$, $J_{osmo}$, and $J_{phago}$ for each generalist size bin, while the specific trophic strategy is not explicitly calculated. The

observations of autotrophic organisms in the CCE-LTER and HOT datasets are on the other hand based on the presence of chlorophyll-*a* in epifluorescence microscopy as well as on DNA and photosynthetic pigments in flow cytometry. In these types of analysis, an organism is either classified as autotroph or heterotroph with no room for distinguishing degrees of mixotrophy. It therefore requires a post-processing of our model result to be able to compare with observations. Our processing approach is described below. To minimize the significance of the uncertain distinction of mixotrophy in

comparison with observations, we also calculate the total biomass (heterotrophic plus autotrophic carbon) of the pico- and nano-sized classes (Fig. 2 e-f). The addition of the heterotrophic component increases the overall biomass of pico- and nanoplankton, especially in the CCE-LTER observations, but has very little influence on the overall size-distribution of the plankton. Finally, we do not calculate the total biomass in the micro-sized bin, as observations in this size class are

significantly underestimated (Taylor et al., 2011). Taylor et al. (2011) finds that micro-sized heterotrophic ciliates are poorly
preserved in the epifluorescence slide-making protocol.

### 3.2 Evaluation metrics

The model result size spectrum is recalculated into different pools of biomass carbon (Table 3): The sum of heterotrophic and autotrophic biomass size classes is referred to as total picoplankton ($B_{T\text{-pico}}$) and total nanoplankton ($B_{T\text{-nano}}$). The autotrophic biomass in the size classes is referred to as autotrophic picoplankton ($B_{A\text{-pico}}$), autotrophic nanoplankton ($B_{A\text{-nano}}$)
and autotrophic microplankton ($B_{A\text{-micro}}$). These different biomass classes are calculated for each autotrophic biomass bins (AC-bins) the same way that Taylor and Landry (2018) processed they observations (Fig. 2).

Table 3: Notation used for different biomass size classes

| Notation | Biomass class | Size range |
|---|---|---|
| $B_{T\text{-pico}}$ | Biomass of total picoplankton | <2 μm |
| $B_{T\text{-nano}}$ | Biomass of total nanoplankton | 2-20 μm |
| $B_{A\text{-pico}}$ | Biomass of autotrophic picoplankton | <2 μm |
| $B_{A\text{-nano}}$ | Biomass of autotrophic nanoplankton | 2-20 μm |
| $B_{A\text{-micro}}$ | Biomass of autotrophic microplankton | 20-200 μm |

To calculate how much of the model biomass should be classified as autotrophic we first define two ratios $\gamma_{A:F} = J_{auto}/(J_{auto} + J_{phago})$ and $\gamma_{A:O} = J_{auto}/(J_{auto} + J_{osmo})$, where J's are the different rates of carbon synthesis defined above. If the ratio $\gamma_{A:F}$ is above 0.1, we classify the generalist in that size bin as a fully photoautotrophic organism for comparison with observations (Stoecker et al., 1996; Stukel et al., 2011). We then calculate the autotrophic biomass in that size bin($i$) based on the combined rates of autotrophic and phagotrophic biosynthesis as:

$$B_{auto,i} = B_i \frac{J_{auto,i} + J_{phago,i}}{J_{auto,i} + J_{phago,i} + J_{osmo,i}}$$

If $\gamma_{A:F}$ is below 0.1, we instead define that the generalist in that size bin is both auto- and phagotrophic and the autotrophic biomass is calculated as:

$$B_{auto,i} = B_i \frac{J_{auto,i}}{J_{auto,i} + J_{phago,i} + J_{osmo,i}}$$

The same philosophy is used for the osmotrophic-autotrophic ratios.

While the use of $\gamma_{A:F}$ is inspired by red/green florescence ratio (~0.08) used to partition mixotrophic nanoplankton into functionally phototrophs or heterotrophs in observational datasets (Stukel et al., 2011), we test our results for a range of values (0.1 −0.9) and find that this range does not change our results quantitatively.

In evaluating our model against observations, we use oceanographic statistical practice as described in Taylor (2001). For each of the 14 AC-bins we first calculate the mean and standard deviation (STD) for the model and for the observations over the years of 2004-2011. Based on these means and STDs we then calculate the model versus observation correlation coefficient ($COR_{m-o}$), root-mean square difference ($RMSd_{m-o}$) as well as centered root-mean square error ($cRMS_{m-o}$) for the 14 AC-bins (Table 4). Statistical comparisons are only made between model and observation AC-bins if there are more than two observations in an AC-bin. The model-observation comparison is based on the upper 100 meters of water columns because this increases the total number of observations through the year. Taylor and Landry (2018) evaluated only the upper 30 meters of their observations. Our reanalysis of their data shows no significant change in the observed distribution of pico-, nano-, and microautotrophic organism relative to their results, when we also include observation between 30 and 100 m.

The statistical measures are objective, but we need to define what acceptable model results are. We work with the premises that we cannot expect to have a better fit to the mean observation (mean of 2004-2011) than the year-to-year variation that is observed at the specific site. For each year between 2004-2011 we therefor calculate annual mean and standard deviation for each AC-bin based on the observations ($STD_{ia}$, Table 4). We refer to differences from year to year as the inter-annual variation in observations. We then evaluate correlation coefficient and root-mean-square difference between the annual mean observation and the total mean observation for all 14 AC-bins (abbreviated $COR_{iao}$ and $RMSd_{iao}$, respectively. Notice the difference from the subscripts above). These statistics inform us how much natural variation occurs around the mean observation. The minimum $COR_{iao}$ and maximum $RMSd_{iao}$ of the inter-annual variation is used to determine if a model result is successful ($COR_{iao}$ and $RMSd_{iao}$ values are available in Supplement S2). For example, if the correlation coefficient of the model average versus the observed total mean is higher than the correlation coefficient of the inter-annual variation ($COR_{m-o}$ > $COR_{iao}$) then the model result for a given parameter set is considered successful in terms of correlation coefficient. Ideally, the optimal successful model simultaneously has $COR_{m-o}$ > $COR_{iao}$ and $RMSd_{m-o}$ < $RMSd_{iao}$ for all biomass size categories. As is clear below, no model results fulfill both criteria for all biomass size categories. Instead, we isolate the model results that fit the COR and RMSd criteria for at least 8 out of 10 size categories and has biomass in $AC_{bio}$-bin up to at least 40 $\mu gCl^{-1}$ for CCE and 15 $\mu gCl^{-1}$ for HOT. For the solutions that fulfill these criteria we sort them according to their $COR_{m-o}$ and $RMSd_{m-o}$ and make a visual qualitative assessment in comparison with observations (cf. Fig 2).

**340**    **Table 4: Definitions of biomass metrics and their calculations**

| Metric | Description | Formula |
|--------|-------------|---------|
| $STD_o$[1] | Standard deviation of observed biomass (o) across $AC_{bio}$-bins (N), calculated from the mean biomass ($\bar{o}$) values. | $$std_o = \sqrt{\frac{\sum_{n=1}^{N}(o_n - \bar{o})^2}{N}}$$ |
| $STD_{ia}$[2] | Standard deviation of biomass for a given year ($o_{ia}$) for each $AC_{bio}$-bins (N), showing inter-annual variability. | $$std_{ia} = \sqrt{\frac{\sum_{n=1}^{N}(o_{ia.n} - \bar{o})^2}{N}}$$ |
| $STD_m$ | Standard deviation of modelled biomass (m) across $AC_{bio}$-bins (N), calculated from the mean modelled biomass ($\bar{m}$) values. | $$std_m = \sqrt{\frac{\sum_{n=1}^{N}(m_n - \bar{m})^2}{N}}$$ |
| $COR_{m-o}$ | Correlation coefficient between modelled biomass and mean observed biomass for each $AC_{bio}$-bins (N) | $$COR_{mo} = \frac{\sum_{n=1}^{N}(o_n - \bar{o})(m_n - \bar{m})}{N}\frac{1}{std_o std_m}$$ |
| $COR_{iao}$ | Correlation coefficient between yearly observed biomass and mean observed biomass for each $AC_{bio}$-bins (N) | $$COR_{iao} = \frac{\sum_{n=1}^{N}(o_n - \bar{o})(o_{ia.n} - \bar{o})}{N}\frac{1}{std_o std_{ia}}$$ |
| $cRMS_{m-o}$ | Centered root-mean square difference between modelled biomass and mean observed biomass for each $AC_{bio}$-bins (N) | $$cRMS_{mo} = \sqrt{\frac{\sum_{n=1}^{N}((m_n - \bar{m}) - (o_n - \bar{o}))^2}{N}}$$ |
| $RMSd_{m-o}$ | root-mean square difference between modelled biomass and mean observed biomass for each $AC_{bio}$-bins (N) | $$RMS_{mo} = \sqrt{\frac{\sum_{n=1}^{N}(m_n - o_n)^2}{N}}$$ |
| $RMSd_{iao}$ | root-mean square difference between yearly observed biomass and mean observed biomass for each $AC_{bio}$-bins (N) | $$RMS_{iao} = \sqrt{\frac{\sum_{n=1}^{N}(o_{ia.n} - o_n)^2}{N}}$$ |

[1]*$STD_o$ represents the variability of biomass among size classes within the dataset averaged across the years 2004 to 2011. This is a mean of all data within the upper 100 meters for all sampling data and location and interannual variability is thus not present here.*

[2]*$STD_{ia}$ captures how biomass in each size class for a given year deviates from the dataset averaged across the years 2004 to*
**345**    *2011.*

### 3.3 Initial- and boundary conditions

The analyses are performed in a water-column setup of the NUM model with vertical diffusion and temperature profiles for the two sites extracted from the global 1° transport matrix MITgcm_ECCO (Stammer et al., 2004). Light, expressed as PAR, is modeled according to daily insolation depending on the specific latitude, day of the year, and time of day. The NUM model uses nitrate as nutrient and is initialized with annual mean observations of nitrate concentrations based on data from CCE-LTER and HOT (Calcofi-Scripps-Institution-of-Oceanography and Wilkinson, 2022; Pasulka et al., 2013; Karl and Lukas, 1996) The nitrate observations have been smoothed with a Gaussian filter to reduce noise. The observations from Station ALOHA only include nutrient measurements to a depth of 175 meters. Mean nitrogen values from World Ocean Atlas 2018 are used below this depth (Garcia, 2018; Garcia et al., 2019).

The model is simulated with 10 logarithmically distributed size classes of generalists in the range from 3 pgC to 10 μgC, equivalent to a spherical cell diameter of approximately 0.25 μm to 363 μm. In addition to the 10 size classes of generalists, the model has small and large detritus of particulate organic carbon with different sinking velocities. The model is run for 15 years with daily output. The last five years are averaged and evaluated to smooth out inter-annual differences in model results. DOC is initialized with a value of 60 μmol kg$^{-1}$ (Zakem and Levine, 2019; Sarmiento and Gruber, 2006; Letscher and Moore, 2015). DOC rapidly decreases to dynamic steady state with an annual mean value of $\sim 1 \pm 0.5$ μmol kg$^{-1}$.

## 4 Results

### 4.1 Model validation

Initial simulations have shown that 15 years is sufficient to produce a dynamic steady state with steady annual cyclicity. Of the 100,000 simulations for CCE, less than 1% terminated due to instability generated by the combinations of parameters. Random sampling of the simulations that integrated properly (completed) showed results were reproducible in re-runs and that the model had reached dynamic steady state.

To validate the model's first-order response, we simulated conditions for the California Current Ecosystem (CCE) and Station ALOHA using the reference parameters from Table 1. The results were then compared to observed biomass spectra and nitrogen depth profiles for the two sites (Fig. 3). The contrasting oceanographic regimes between the sites are evident from their nitrogen profiles (Fig. 3a, b). The California Current Ecosystem, characterized by coastal upwelling, shows a nitricline at approximately 100 meters depth. In contrast, Station ALOHA, an oligotrophic open-ocean site with downwelling, exhibits lower nitrogen levels and a deeper nitricline. The model responds correctly to the difference in circulation at the two sites resulting in higher nitrogen concentration at CCE compared to Station ALOHA. Although the model's results generally align with observations, there is a depressed nitricline at CCE, leading to lower-than-expected

nitrogen values in the upper 200 meters of the water column and slightly elevated nitrogen concentrations at Station ALOHA.

Despite these differences in nitrogen profiles, the biomass size distributions at both sites are remarkably similar (Fig. 3c, d). Both sites display a relatively flat Sheldon biomass spectrum, with a mean biomass of approximately 1 µgC/L at CCE and approximately 0.5 µgC/L at Station ALOHA. These biomasses are within the expected range of observations, although the mean observed biomass is slightly higher, averaging 1.5 µgC/L at CCE and 0.5 µgC/L at Station ALOHA. Notably, the largest discrepancy between the model and observations occurs in the small size classes at Station ALOHA, where the model underestimates the biomass. The larger standard deviations observed at CCE indicate a more variable environment compared to Station ALOHA.

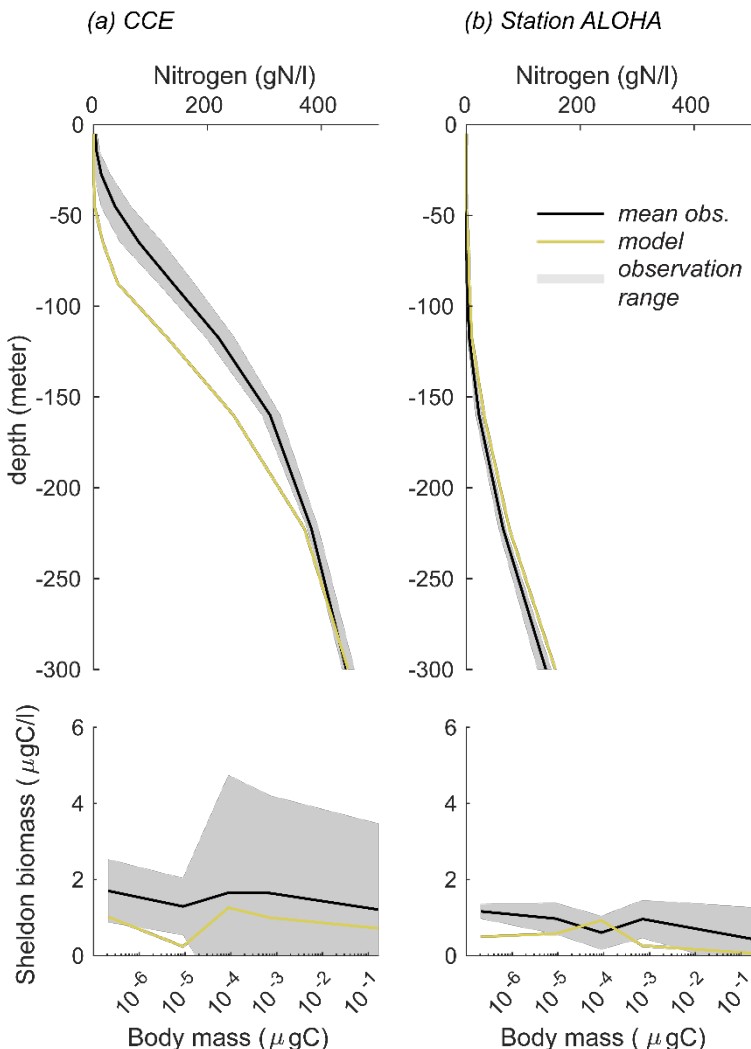

**Figure 3: Nutrient profile and Sheldon biomass comparison between observations and reference model simulation for (a) CCE and (b) Station ALOHA. The Sheldon biomass spectrum illustrates the biomass in each body mass bin normalized with bin width. The Sheldon biomass spectrum is defined as $B_s(m) = B_i / log(\frac{m_i^+}{m_i^-})$ (Andersen and Visser, 2023).**

In the following, we describe the results of the first-level randomized parameter studies and the subsequent detailed studies. The shared aim of these investigations is to better understand NUM model behavior, performance relative to observations, and of how much parameter choice influences model results.

## 4.2 First-level random parameter study: can the model reproduce the planktonic community biomass structure?

We initially tested the model's ability to reproduce the biomass spectrum and the community size spectrum of the California Current Ecosystem (CCE). Just as important, this is also a test of the variability that parameter choices have on the model result. The result of the simulations is illustrated in Taylor diagrams (Fig. 4). The Taylor diagrams provide a visual representation of the normalized standard deviation ($STD_m/STD_o$; radial distance from origin shown as grey solid line), correlation ($COR_{m-o}$; azimuthal positions) and the centered root-mean-square difference ($cRMS_{m-o}$; black circles extending out from the grey dot) of the 100,000 model simulations, compared to the annual averaged observations from CCE (represented by the grey dot). The bright yellow color in the first quadrant of all five diagrams show that the model simulations generally result in a positive correlation coefficient with the CCE-LTER observations on all biomass size categories. The smallest effect of parameter variations is seen om the autotrophic microplankton ($B_{A-micro}$, Fig4e) where solutions are centered in a smaller area than the other four size categories. On the other end of the scale autotrophic picoplankton show the larges spread in solutions from randomizing the parameters ($B_{A-pico}$, Fig4c). On average, the smallest difference between model result and the mean observations (determined as $abs(1 - mean(cRMS))$) is found for $B_{T-nano}$ which, despite some simulations with a negative correlation, generally show the closest fit to observation. The other four categories have a larger deviation from observations due to either lower pattern agreement ($COR_{m-o}$) or over- or underestimation of the amplitude of variations ($STD_m/STD_o$). The amplitude of variation in the size spectrum is overestimated for all size groups of pico- and nanoplankton while the model underestimate the amplitude of variation in the autotrophic micro plankton. The pattern agreement is overall best for $B_{T-pico}$ and $B_{A-nano}$ with mean correlation of 0.87 and 0.80. Interestingly, the result of the simulations falls within three distinct groups for $B_{A-pico}$, where some parameter combinations produce a much better correlation with observations than others. That $B_{A-pico}$ fall in three groups may be related to the biomass quantization also found in observations and other size-structured planktonic ecosystem models (Moscoso et al., 2022; Schartau et al., 2010).

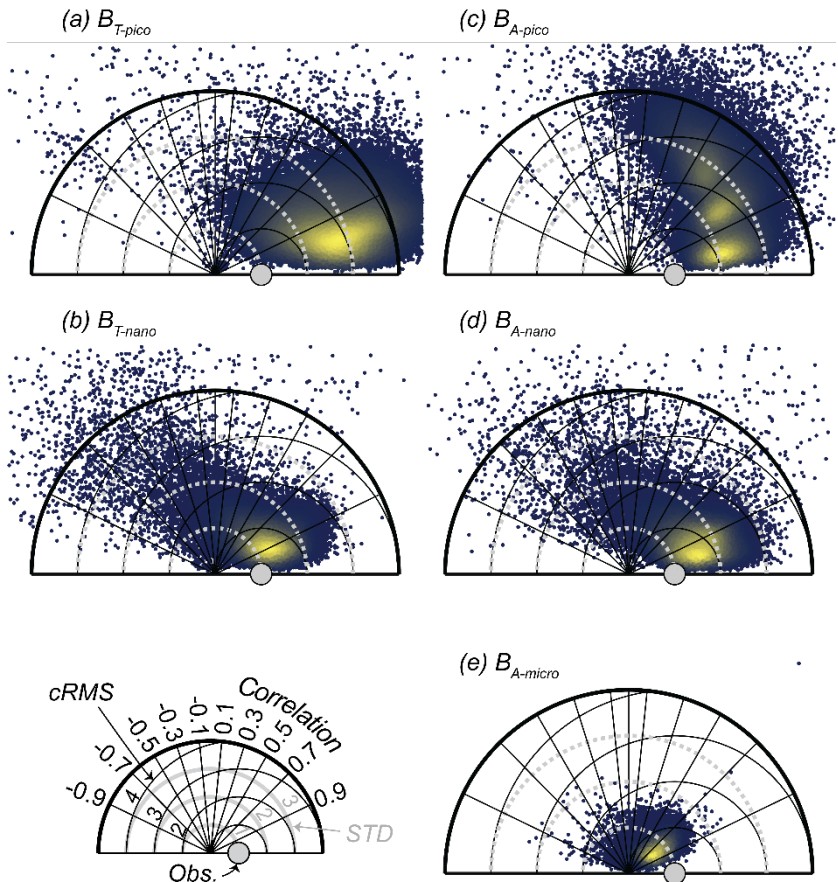

**Figure 4: Taylor diagrams for 100,000 random parameter combinations at CCE, displaying the standard deviation (STD) of the model result relative to observations (Obs.), and correlation coefficient (COR), and centered root-mean-square-difference (cRMS) between model and observations. Blue to yellow color reflect increasing number of realizations in each area. $B_T$ and $B_A$ are defined in figure 2.**

An alternative way to get an overview of the model's capabilities and parameter effect is to visualize of the overall trend in simulations compared to the observation data as a density plot (Fig. 5). The coloring on the figure show that most of the simulations for the five size categories have a power-law increase of the biomass with increasing AC-bins. Generally, the model does not capture the occurrences of high $AC_{bio}$ concentrations ($AC_{bio}$ above approximately 100 µg C/l), which is consistent with the observation that only 2% of the samples have autotrophic biomass concentrations of 100 µg C/l or above, here illustrated by the size of the white dots. The trend in simulations corresponds relatively well with observations for $B_{T-nano}$ (compare to mean observations given as white dots) which also showed to be the size category with the lowest cRMS

(fig. 4). The trend in $B_{A-nano}$ simulations also aligns reasonably well with observations, though the correlation is slightly
weaker due to a larger discrepancy between the modelled increase in biomass and the observed increase in biomass. Both

size groups of nanoplankton do however underestimate the biomass at low $AC_{bio}$-bin ($AC_{bio}< 10$ μgCl$^{-1}$) and overestimate

the biomass at higher $AC_{bio}$, corresponding with the greater-than-observed amplitude of variations in the Taylor diagrams.

The picoplanktonic size groups also exhibit a gradual increase in biomass with increasing $AC_{bio}$, rather than the plateau at

intermediate-high $AC_{bio}$ seen for observations of $B_{T-pico}$ and the decrease in biomass for $B_{A-pico}$. Additionally, the model

underestimates biomass at low $AC_{bio}$ for both picoplanktonic size groups. The modelled amplitude of variation for $B_{A-micro}$ is

lower than the observations which manifest as a too high biomass at low $AC_{bio}$ and a lower-than-observed increase in

biomass with increasing $AC_{bio}$.

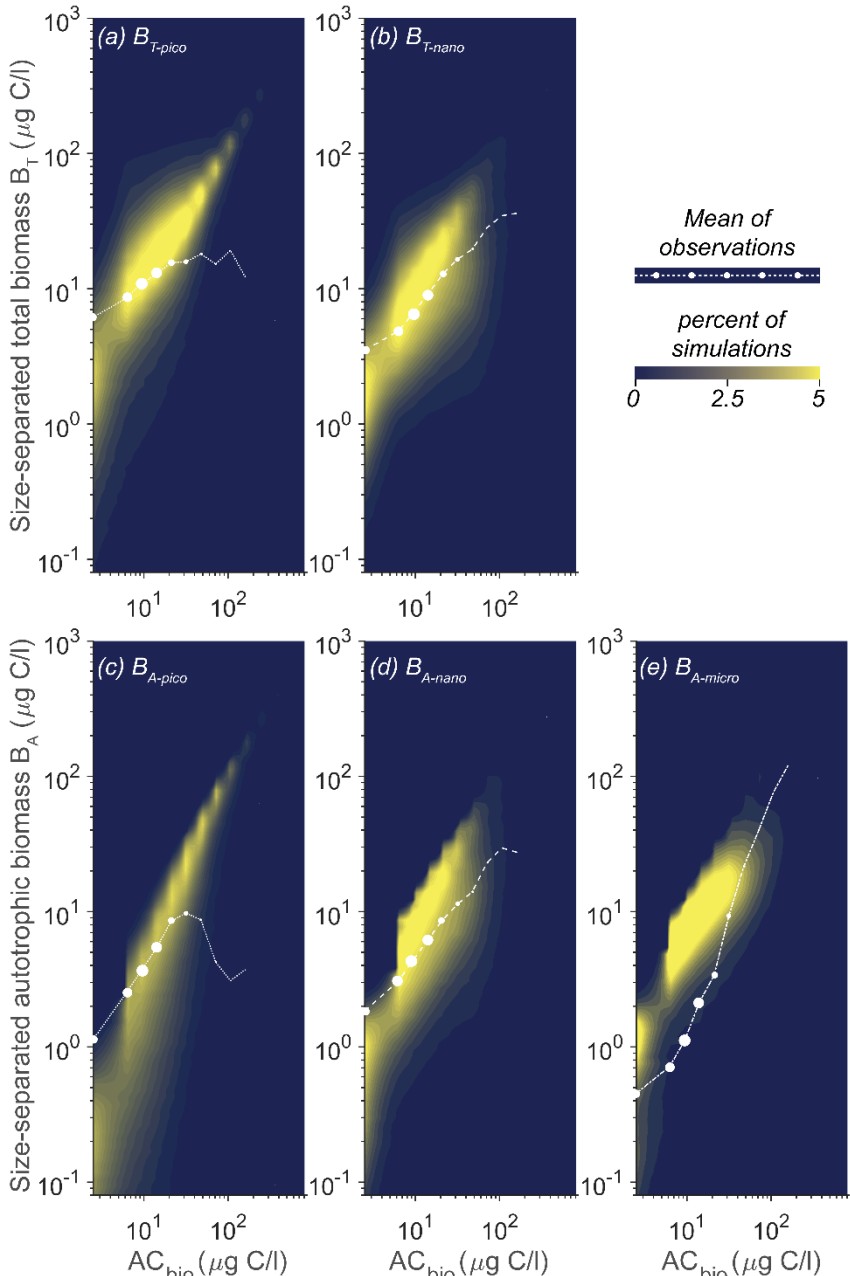

**Figure 5: Model mean and total biomass of size groups as a function of total biomass for 100,000 random parameter combinations at CCE. White dots are observations in AC$_{bio}$ bins. Abbreviations as figure 2. Blue to yellow reflect an increasing number of realizations in each area. The size of white observations dots indicates the relative number of observations in that AC$_{bio}$ bin. Note the tendency of NUM to under-estimate pico- (a, c) and nanoplanktonic (b, d) biomass at low AC$_{bio}$ while overestimating the biomass at intermediate-high AC$_{bio}$. The autotrophic microplanktonic biomass (e) is generally overestimated.**

Of the completed model calculations, the ideal parameter combinations should result in $COR_{m-o} > COR_{iao}$ and $RMSd_{m-o} <$ $RMSd_{iao}$ for all size groups. Evaluating these conditions shows that none of model results fulfill both criteria for all size groups. A detailed examination of the simulations in term of $COR_{m-o}$ and $RMSd_{m-o}$, however, reveals a subset of seven simulations that result in a planktonic size variability that corresponds particularly well with the observations (Fig. 6).

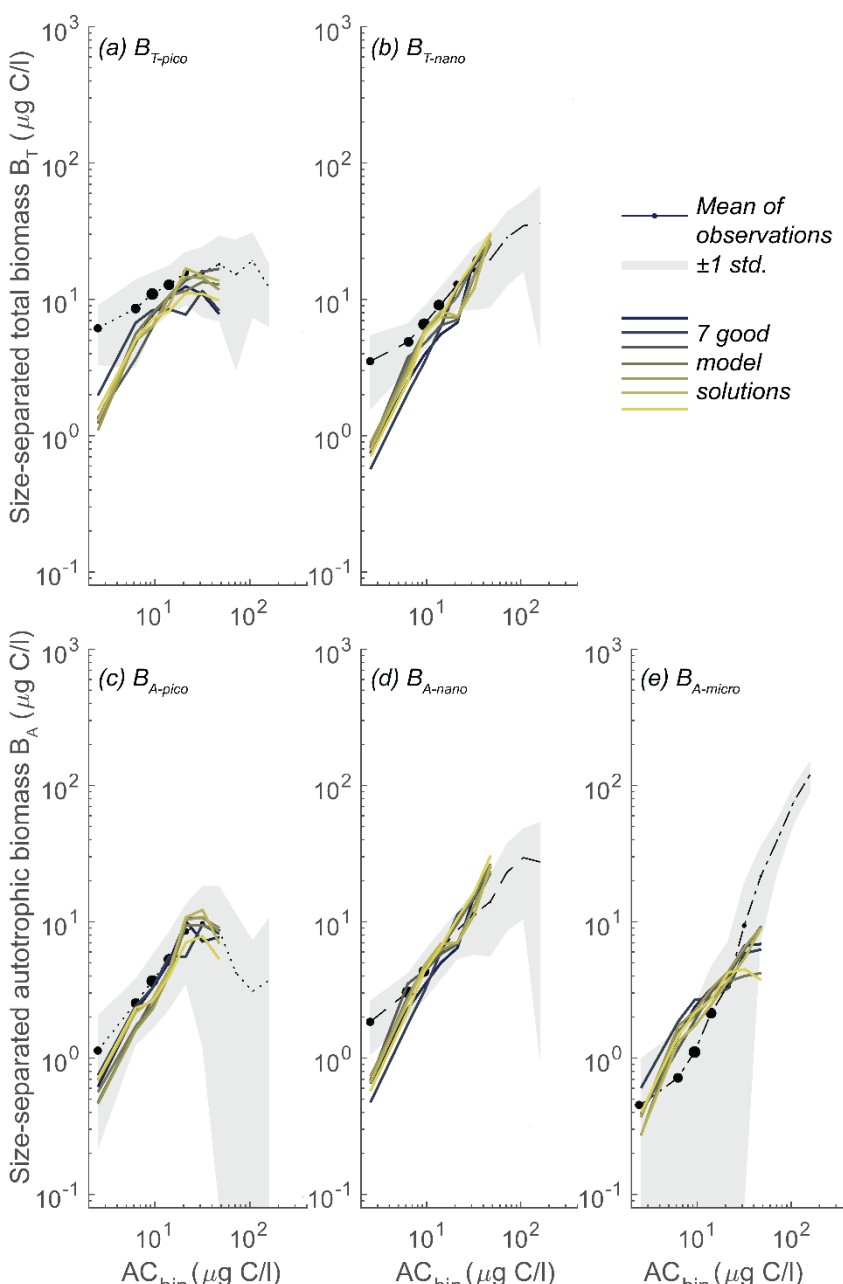

**Figure 6: Model mean and total biomass of size groups as a function of total biomass for the 7 statistically most optimal parameter combinations at CCE. Black dots are observations in $AC_{bio}$ bins. Abbreviations as Fig. 2. Note the tendency for NUM to underestimate pico- and nanoautotrophic biomass at very low $AC_{bio}$ and overestimate microautotrophic biomass.**

In these seven simulations the picoplanktonic size groups align with the mean observations, showing a plateau at intermediate-high $AC_{bio}$ for $B_{T-pico}$ and a tendency for biomass to decrease for $B_{A-pico}$ at $AC_{bio}$ above 30 µgCl⁻¹ (Fig. 6a, c). The parameter combinations do however result in $B_{T-pico}$ lower than one standard deviation for the smallest $AC_{bio}$ bin. Both nanoplanktonic size groups closely follow the observations, though still with lower-than-observed biomass at low $AC_{bio}$ (Fig. 5b, d). The trend in microautotrophic biomass aligns with most of the model results, which generally show higher-than-observed biomass. These results fall on the lower end of the 100,000 simulations but are still too high at low to intermediate autotrophic biomass levels (approximately 4-30 µgC/L), forming a "humped back" shape (Fig. 5e). While these seven simulations correlate remarkably well with the observations, the seven simulations general have slightly too low correlation coefficient for $B_{A-micro}$ (0.79-0.95 for model results versus 0.96 for observations) and too high root-mean-square-difference for $B_{T-pico}$ (0.48-0.64 versus 0.4) and $B_{T-nano}$ (0.37-0.71 versus 0.3) (the statistic is available in Supplement S3).

With the goal of identifying a parameter range that yields robust optimal solutions, we will focus on this subset of seven simulations that perform especially well in the further sensitivity analyses of the parameters. We use the identified seven sets of parameters to define a *restricted parameter span* based on the on minimum and maximum of each parameter in the set group (Fig. 7, blue lines).

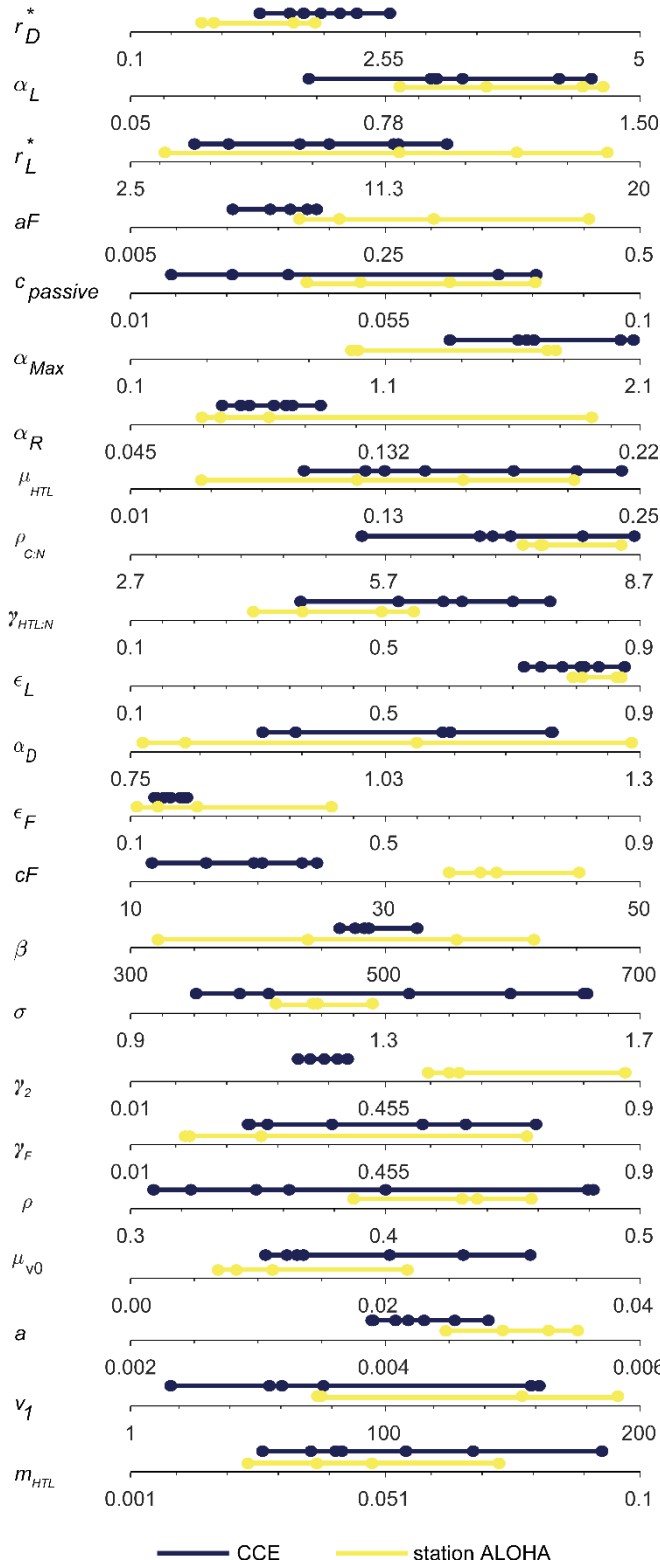

$r_D^*$

0.1    2.55    5

$\alpha_L$

0.05    0.78    1.50

$r_L^*$

2.5    11.3    20

$aF$

0.005    0.25    0.5

$c_{passive}$

0.01    0.055    0.1

$\alpha_{Max}$

0.1    1.1    2.1

$\alpha_R$

0.045    0.132    0.22

$\mu_{HTL}$

0.01    0.13    0.25

$\rho_{C:N}$

2.7    5.7    8.7

$\gamma_{HTL:N}$

0.1    0.5    0.9

$\epsilon_L$

0.1    0.5    0.9

$\alpha_D$

0.75    1.03    1.3

$\epsilon_F$

0.1    0.5    0.9

$cF$

10    30    50

$\beta$

300    500    700

$\sigma$

0.9    1.3    1.7

$\gamma_2$

0.01    0.455    0.9

$\gamma_F$

0.01    0.455    0.9

$\rho$

0.3    0.4    0.5

$\mu_{v0}$

0.00    0.02    0.04

$a$

0.002    0.004    0.006

$v_1$

1    100    200

$m_{HTL}$

0.001    0.051    0.1

CCE          station ALOHA

 **Figure 7: Span of all free model parameters for the seven most optimal parameter combinations at either CCE (blue) or Station ALOHA (yellow). The seven CCE model results are shown in Fig. 6. Note how the parameter spans for CCE and Station ALOHA generally follows the same trend except for $\gamma_2$ and cF where higher values are needed to fit the data at Station ALOHA than CCE. The most optimal parameter combination is estimated by highest correlation coefficient and lowest root-mean-square difference between model and observation simultaneously for $B_{T\text{-pico}}$, $B_{T\text{-nano}}$, $B_{A\text{-pico}}$ $B_{A\text{-nano}}$, and $B_{A\text{-micro}}$. Parameter definitions in Table 1 and**
**other abbreviations in Fig. 2.**

## 4.3 Restricted parameter span and sensitivity

We will now aim to evaluate the importance of the parameter uncertainties and to establish a stable parameter space for the 23 free parameters, wherein the simulations yield a reasonable result. The range of each free parameter is based on the range defined by the seven solutions with optimal fit (Fig. 6). An initial local parameter sensitivity assessment revealed a high
degree of non-linearity in the model that makes it difficult to make any global conclusions about parameter influence based on local studies. To gain more insight into how the parameters influence the sensitivity of the entire non-linier ecosystem system we instead perform a global sensitivity analysis (Bilal, 2014; Sobol, 2001; Zhou and Lin, 2008).

Figure 8 displays the parameters ranked by Sobol's total sensitivity index (STi) based on root-mean-squared difference for the five size groups. The corresponding figure based on correlation is available in Supplement S6, but its conclusions are
consistent with Fig. 8. The value of the index cannot be compared across the different categories but the span in values gives an indication of the variability in the sensitivity across parameters. For example, while $B_{T\text{-pico}}$ seems to be especially sensitivity to approximately half of the parameters there is little difference among the parameter sensitivity for $B_{A\text{-micro}}$. The global sensitivity analysis reveal that all size groups are sensitive to the choice in parameters that control mortality (red dots): phagotrophy (the phagotrophic assimilation rate ($\varepsilon_F$), clearance rate (aF), the predator-prey ratio ($\beta$) and width ($\sigma$)),
higher trophic level mortality (HTL pressure ($\mu_{HTL}$)), and viral lysis ($\mu_{v0}$). All size groups are also sensitive to the value of maximum rate of biosynthesis ($\alpha_{max}$) and to a smaller degree respiration ($\alpha_R$) (grey dots). Parameters such as the remineralization rate of dead organisms ($\gamma_2$, purple), diffusive affinity cross-over ($r^*_D$, except for $B_{A\text{-micro}}$, blue), and the C:N ratio of the cell are among the moderately sensitive parameters. The parameters mentioned above are parameters that control the predation pressure, biosynthesis and nutrient availability and uptake. Finally, the analysis shows that the picoautotrophic
biomass is more sensitive to the light uptake efficiency ($\varepsilon_L$, yellow) parameter than the other size groups. The analysis shows that other parameters are less important and therefore allows for larger uncertainties. These parameters include the carbon density of the cell ($\rho$), passive losses coefficient ($c_{passive}$), and the remineralization of feeding losses ($\gamma_F$) and higher trophic levels ($\gamma_{HTL}$).

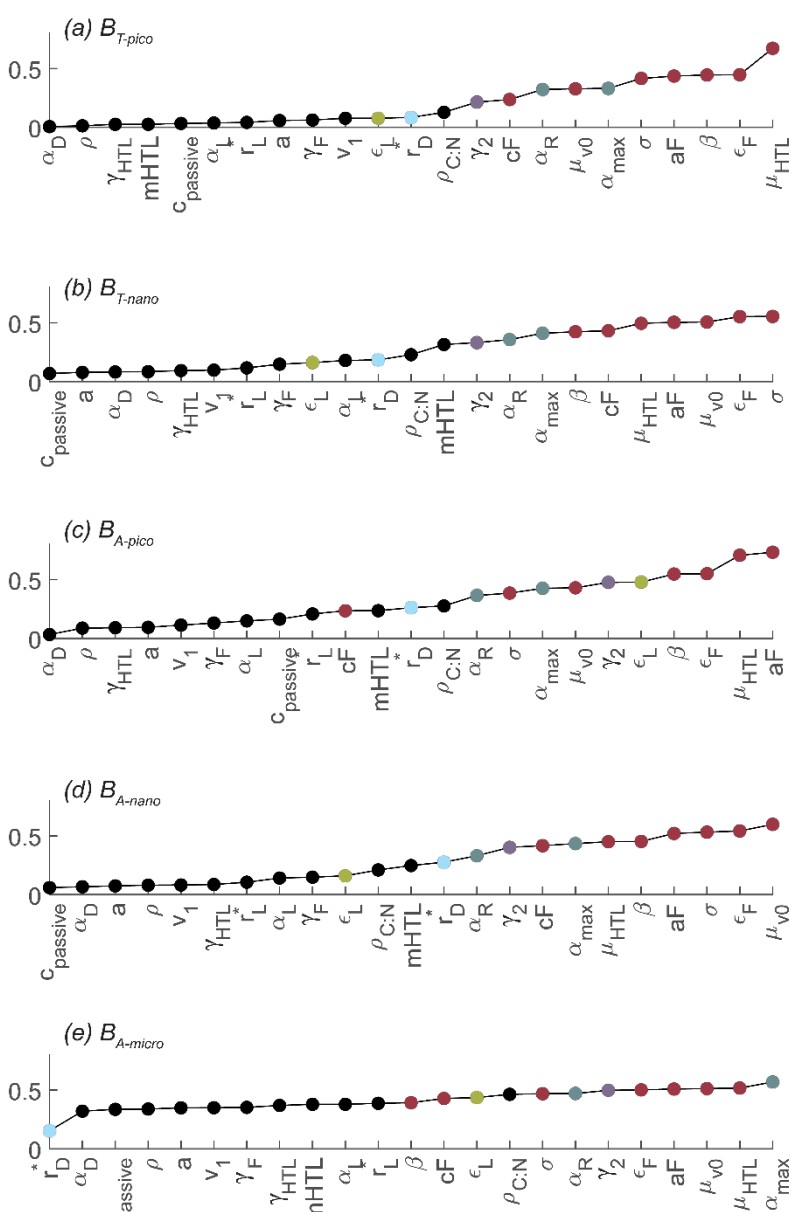


**Figure 8: Global parameter sensitivity ranked based on sensitivity index calculated by Sobol's variance-based sensitivity method for non-linear models. Sensitivity calculated for RMSd. The parameters that the NUM model is most sensitive to have been colored according to categories; predation and mortality (red), synthesis (grey), cell remineralization (purple), light uptake efficiency (yellow), diffusive affinity cross-over (blue). Note how all biomass size is especially sensitive to parameters controlling predation**

**(red dots), and synthesis (grey dots). Parameter definitions in Table 1 and other abbreviations in Fig. 2**


While the model sensitivity towards parameters is complex and non-linear, a final set of 10,000 random parameter simulations demonstrate that the model result space can be reduced by confining the parameter space to the restricted parameter span found in Fig. 7. Using the restricted parameter span we see a tighter fit of model results to observations (Fig. 9) in contrast to the full randomized parameter spans (Fig. 5). The parameter restrictions have removed the simulations that

produced excess pico- and nanosized biomass at high $AC_{bio}$ and the simulations now follow the observed trend with an onset of a plateau at $AC_{bio}$ of 20 $\mu gCl^{-1}$ for $B_{T\text{-}pico}$ and $B_{A\text{-}pico}$. The restriction has had less impact on the nanoplanktonic biomass but has narrowed the range of results, leading to a slight overestimation of nanoplanktonic biomass in most simulations, particularly at $AC_{bio}$ levels above 30 $\mu gCl^{-1}$. Overall, the model results in Fig. 9 demonstrate a notable improvement in model performance for the identified parameter spans in comparison to the full parameter space. While this improvement may seem

intuitive it is not necessarily a priori given, considering the model's parameters non-linear response to parameter change. The local sensitivity analysis showed that, even within the restricted parameter space, the impact of varying a parameter is highly dependent on the other parameters (Fig. S2). The restricted parameter space could therefore, in theory, have resulted in the same degree of model misfits as the full parameter span with only a few acceptable solutions generated by very specific parameter combinations. That the model performance is enhanced by restricting the parameter span suggests that

further detailed parameter tuning may not be necessary to achieve reliable results from the NUM model. While a better performance is encouraging it is important to evaluate if the identified parameter spans are applicable to other biogeographic provinces.

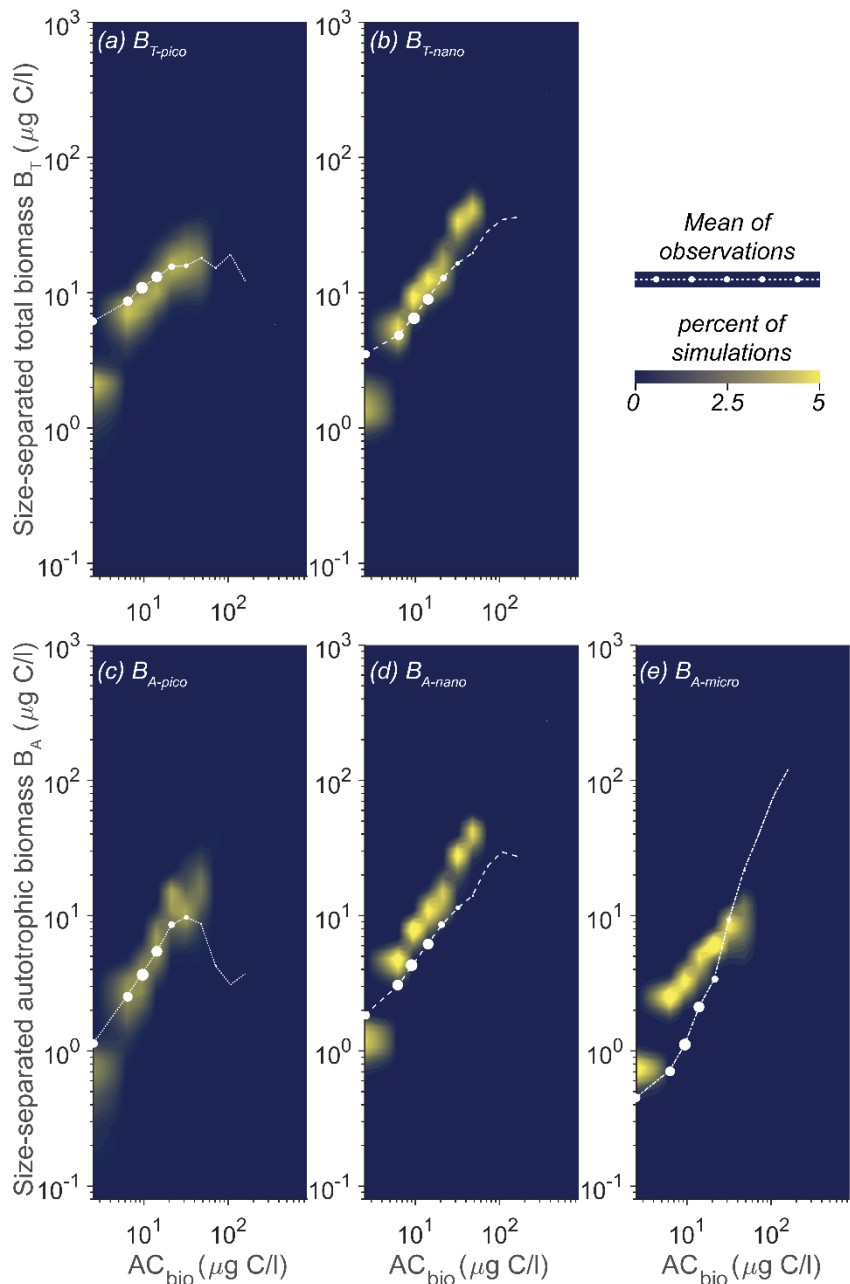

Figure 9: Model mean and total biomass of size groups as a function of total biomass for 10,000 random parameter combinations sampled within the restricted span of parameters at CCE. The random parameter spans are based on the parameter range of the seven statistically optimal parameter combinations at CCE (see text). White dots are observations in $AC_{bio}$ bins. Abbreviations as Fig. 2. Blue to yellow reflect an increasing number of realizations in each area. Note how the solution space has been restricted, especially for picoplankton (a ,c) compared to the full parameter span (Fig. 5).

## 4.4 Results for Station ALOHA

The heart of the trait-based approach is its potential universality; the principle that a single set of parameters can describe organisms and ecosystems across time and place. An important test is therefore if the parameter sets that performed best at CCE are suited for different oceanographic settings. Figure 10 shows the result of 10,000 simulations with conditions mimicking Station ALOHA with quasi-random parameters from within the restricted parameter span defined for CCE. The model reacts to the shift in oceanographic regime by lowering the overall autotrophic biomass. Most simulations only reach a biomass of 20 µg C l$^{-1}$ which is consistent with observations. The biomass of the picoplanktonic size groups is lower than mean observation but generally within one standard deviation (Fig 10 a, c and see Fig. 2b for comparison). Both nanoplanktonic size groups exhibit elevated biomass relative to observations, with the discrepancy being larger than that observed for the nanoplanktonic size groups in the CCE simulations (Fig 10b, c, compare with Fig 9b, c). The microautotrophic size group exhibits the poorest correlation with mean observations, displaying excessive biomass at low $AC_{bin}$ (< 9 µg C l$^{-1}$) and variable, but lower biomass at $AC_{bin}$ above 9 µg C l$^{-1}$. This pattern is inconsistent with the observed sigmoidal trend, although the biomass falls within the standard deviation of the observations (Fig 10e, compare with Fig. 2b). A comparison to the first-level random parameter simulation with 100,000 simulations within the full parameter space (not illustrated here but available in Supplement S7) show that restricting the parameters based on the solutions from CCE has removed a set of simulations that produced too large biomasses for all size categories at intermediate AC-bins. However, it also eliminates a set of simulations with better fitting biomass concentrations at low $AC_{bio}$-bins. Figure 11 shows a set of simulations from the first-level random parameter study that performs particularly well for Station ALOHA. In these simulations, both pico- and nanoplankton follow the trend of the observations and exhibit the correct amount of biomass. Interestingly, most of the parameters for these four simulations fully overlap with parameter for the best solutions at CCE (Fig. 7). While some parameters such as aF and $\alpha_{max}$ only partly overlap, the only parameters that significantly differs between the two sites are $\gamma_2$, cF that both have higher value at Station ALOHA than CCE. The parameter $\gamma_2$ controls the fraction of dead matter directly remineralizer back to nutrients and it is thereby an important parameter in controlling the amount of osmotrophy for the smallest planktonic size group. cF and aF are two important components of the rate of phagotrophy. It is noteworthy that the parameters for successful solutions at the two different sites exhibit parameter trends in many cases correlate; for both stations, the successful simulations have relatively high $\alpha_L$, $\alpha_{max}$, $\rho_{C:N}$, $\varepsilon_L$, a, and low $\varepsilon_F$.

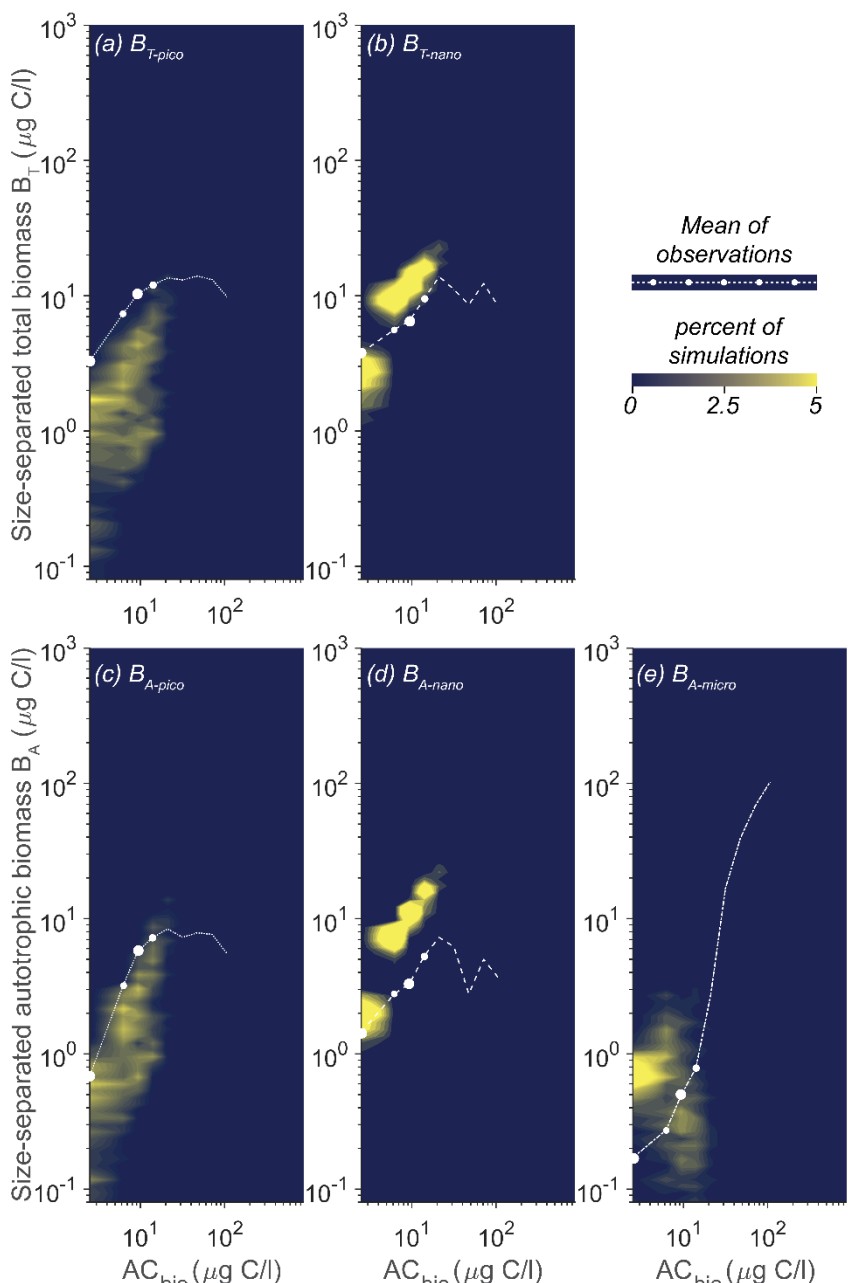

**Figure 10: Model mean and total biomass of size groups as a function of total biomass for 10,000 random parameter combinations at Station ALOHA. The simulations have random parameter combinations within the restricted parameter space based on the successful simulations from CCE (Fig. 7). White dots are observations in $AC_{bio}$ bins. Abbreviations as Fig. 2. Blue to yellow color reflect increasing number of realizations in a given area. Note how biomass of pico plankton (a, c) is underestimated while nano**

plankton (b, d) is generally overestimated. Microautotrophic plankton (e) has the lowest correlation of the five size classes with decreasing biomass as a function of $AC_{bio}$.

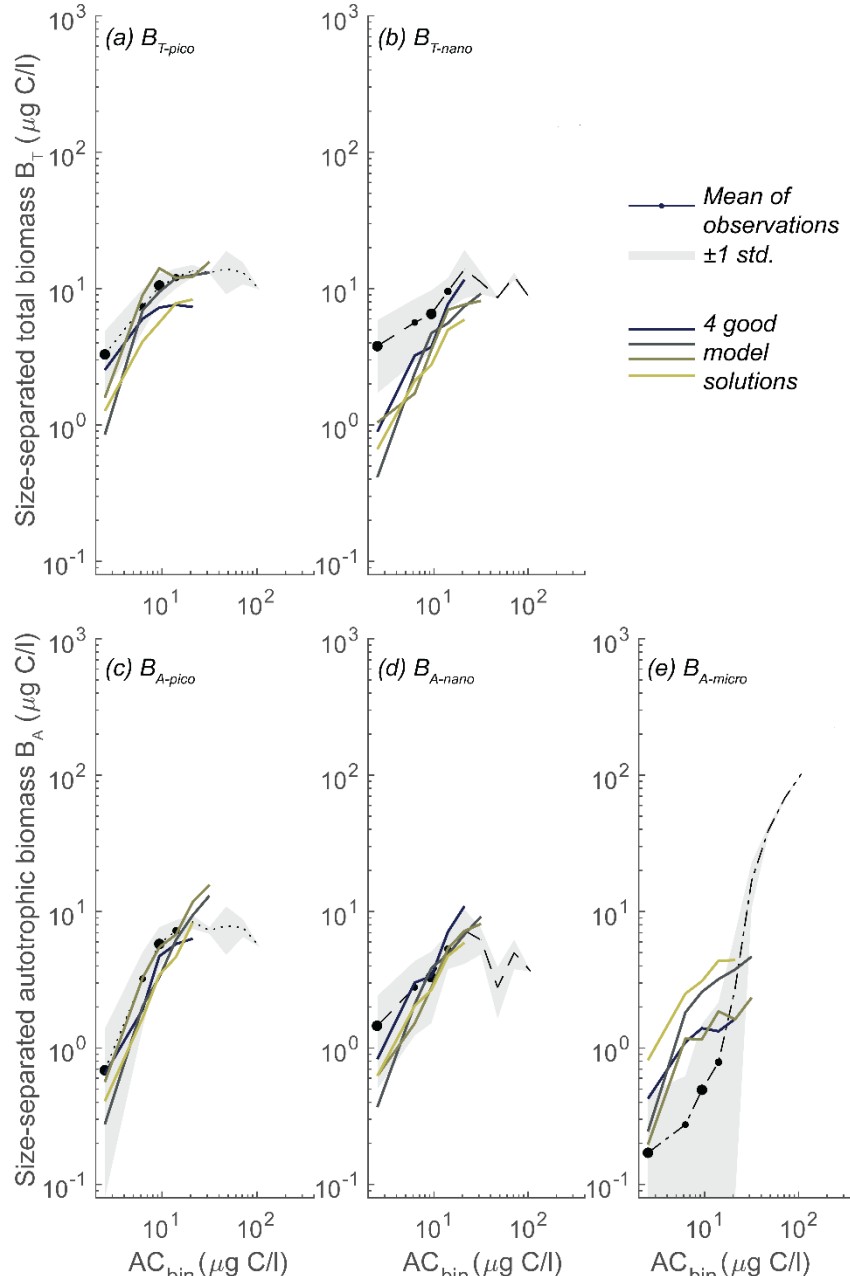

**Figure 11: Model mean and total biomass of size groups as a function of total biomass for the four statistically most optimal parameter combinations at Station ALOHA. Black dots are observations in $AC_{bio}$ bins. Abbreviations as Fig. 2.**

## 4.5 Nutrient profiles

As an indirect way of evaluating the model performance and response to the different environmental conditions, we also evaluate the depth profile of model nutrients for the two sites (Fig. 12). Importantly, the nutrient profile was not part of the initial statistical measures used to identify the model parameters. The nutrient profiles for CCE are remarkably consistent across the solutions. Nutrient concentrations are low in the upper photic zone and increase with depth. While the modelled profiles generally align with the observed data, there is a tendency to underestimate nitrate concentrations at depths ranging between 50 and 200 meters. For Station ALOHA, the modelled profiles also align well with the measured concentrations, with a slight tendency to overestimate nutrient concentrations at depth. The model is generally able to respond correctly to the shift from eutrophic to oligotrophic conditions.

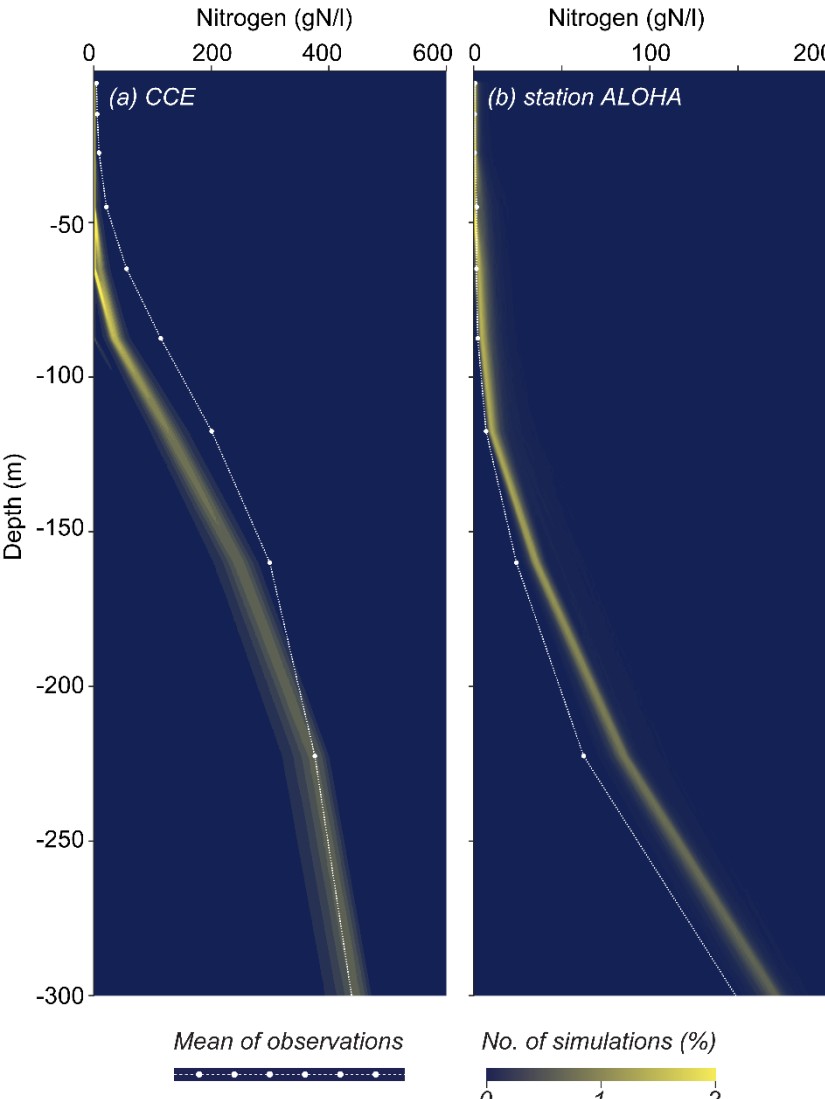

**Figure 12: Model nutrient profiles at CCE (a) and Station ALOHA (b). White dots are observations based on (Calcofi-Scripps-Institution-of-Oceanography and Wilkinson, 2022; Pasulka et al., 2013; Garcia, 2018). Note the model tendency to underestimate N in the thermocline at CCE, but overestimate at Station ALOHA.**

## 5 Discussion and Perspectives

### 5.1 Summary of model performance

We have validated the generalist unicellular NUM ecosystem model toward two quite different biogeographic provinces: the high productivity upwelling conditions of the California Current Ecosystem and the oligotrophic downwelling conditions

at Station ALOHA. For the California Current Ecosystem, out of 100,000 random combinations of the 23 free parameters a large majority of the model results have correlation coefficient toward observations ($COR_{m-o}$) better than 0.7. This demonstrated that the generalist unicellular NUM model, despite its simplicity, can capture the size distribution of the planktonic ecosystem and its nutrient profile over a broad range of parameter values. Out of the random simulations we find only seven optimal, but quite different, parameter combinations that reproduce results for the CCE. These seven optimal simulations almost perfectly match the distribution of each of the size groups as a function of increasing $AC_{bio}$ (Fig. 6). The seven optimal parameter combination have mean $COR_{m-o}$ of 0.94 and $RMSd_{m-o}$ 0.4 for the five size groups considered in comparison with observations. In particular, we find that $B_{A-pico}$ peak and $B_{T-pico}$ plateau at intermediate levels of autotrophic biomass in agreement with observations (Taylor and Landry, 2018) (Fig. 6a,c). We also find a power-law increase in $B_{T-nano}$ and $B_{A-nano}$ as function of $AC_{bio}$ as in observations (Fig. 6b, d). Finally, we observe a "humped back" increase in $B_{A-micro}$ that has the lowest correlation to observations but still within one standard deviation of the observed total mean (Fig. 6e).

Moving to oligotrophic ALOHA station, we find that the seven optimal model parameter combinations from CEE give model results that capture many important aspects of the observational data. NUM qualitatively model a reduction in biomass at Station ALOHA relative to CEE and it generally reproduce the overall size structure. That the NUM model produces less biomass at ALOHA is consistent with observational differences between CEE and ALOHA (Taylor and Landry, 2018). The seven simulations do however produce too low picoplanktonic biomasses and too high nanoplankton biomasses, compared to the observations. A detailed analysis shows that another set of parameter combinations were better at reproducing the pico- and nanoplanktonic biomass both in term of correlation and overall biomass values. The parameter space for these simulations were only significantly different from the restricted parameter span for CCE in their range of a few parameters (discussed below). Our validation against ALOHA overall indicates that by restricting the parameters, based on the seven optimal models at CCE, and focusing on this small set of parameters, it is possible to match the overall increase and decrease in biomass for the different size classes to a degree that would be satisfactory for applications where site-by-site calibration is not possible.

## 5.2 Parameter sensitivity

Our sensitivity analysis shows that the model parameter sensitivity is dependent on the specific parameter combinations and that the ecosystem response is non-linier. Local sensitivity analysis revealed that while one of the good solutions was nearly equally sensitive to almost all parameters, another was mainly sensitive to only one parameter ($\varepsilon F$, Fig S2 in Supplement 4). Through a global sensitivity analysis, we identified the parameters that are especially controlling (Fig. 8). Parameters regulating predation and mortality, biosynthesis, and respiration are generally important among all size groups. Changes in these parameters create the largest shifts in the model output. Interestingly, many of the parameters that produce the largest shifts in biomass are also among the least constrained (Table 1; cf. (Andersen and Visser, 2023)). In the following discussion we focus on the parameters that are the least constrained while also resulting in the largest sensitivity. Higher trophic level mortality ($\mu_{htl}$) is important for all size groups. $\mu_{htl}$ is an extrinsic parameter that governs predation rates by

higher trophic levels. This parameter serves as a closure term in the model and plays a critical role in shaping biomass distribution. Specifically, $\mu_{htl}$ determines the size and biomass of microplankton, initiating cascading effects on smaller size classes. While $\mu_{htl}$ significantly impacts biomass partitioning across size groups, its influence on total biomass is limited

because reductions in microplankton results in corresponding increases in nanoplankton (see Fig. 15b in  Andersen and Visser (2023)). The value of $\mu_{htl}$ depends on the biomass and efficiency of higher trophic levels, which can vary significantly between eutrophic and oligotrophic environments. Our results indicate that optimal $\mu_{htl}$  is larger at CCE compared to Station ALOHA, although there is a significant overlap (Fig. 7). The importance of $\mu_{htl}$ suggests that including higher trophic levels, such as copepods, could reduce model uncertainties. However, that only shifts the problem towards determining the higher

trophic level mortality on copepods, which is equally uncertain. Another highly uncertain parameter that creates large shift in the biomass distribution is the viral lysis mortality coefficient $\mu_{v0}$. This parameter introduces a density dependent control of the population in each size group. It has the effect of increasing the mortality on groups with high biomass and prevents all biomass ending up in one or a few size groups. The principle of abundance controlled viral lysis is an important aspect of the "Killing the Winner" principle (Thingstad, 2000; Winter et al., 2010). The default parameter used in the NUM model is

adjusted such that the effect of viral lysis is smaller than other mortalities, to avoid that this process is determining the result, despite that the value of the parameter is largely unknown. Based on the global sensitivity study it is an important future priority to get a better mechanistic understanding of the viral lysis mortality process. Two other important parameters, cF and εF are both involved in heterotrophic phagotrophy, and are partly multiplicative so one is influencing the other (cf. Eq. 3 Table 2). While the assimilation efficiency (εF) is relatively well-constrained the maximum phagotrophic coefficient (cF) is

not. The parameter cF is unique to the NUM model and determines the phagotrophic assimilation limit for large cells. While cF only directly influences the largest cells it causes a cascading effect down the size spectrum. The default value used here is fitted against maximum growth rate for different types of plankton (see Fig. 5 in Andersen and Visser (2023)). Interestingly, cF is one of the only parameters that show significantly different optimal values for CCE and Station ALOHA (11-25 $\mu$g d$^{-1}$ for CCE versus 35-45 $\mu$g d$^{-1}$ for Station ALOHA). The difference is likely related to a tradeoff between food

acquisition and predation, an important aspect of the slow-fast tradeoff (Salguero-Gómez et al., 2016; Kiørboe and Thomas, 2020). High rates of predation induces higher food acquisition but comes with a higher predation risk (Kiørboe and Thomas, 2020). The difference between CCE and Station ALOHA can therefore be seen as a difference between a more stable environment with high population densities (CCE) and varied conditions with strong environmental gradients (Station ALOHA). The same argument is valid for the phagotrophic clearance rate (aF), where the good fit for Station ALOHA has

higher values compared to CCE. The mechanistic argument for phagotrophic clearance rate relates to the fluid dynamics of a beating flagellate cell (Nielsen and Kiørboe, 2021; Andersen and Visser, 2023). This mechanistic underpinning means that the value of aF is relatively well known, however with a scatter of one of magnitude due to difference in flagella arrangements that generates difference in predation risk. Future investigations into patterns of flagella arrangements in different nutrient environments can maybe give some valuable insight into the trade-off between foraging and predation risk.

675  The last highly unknown parameter that can create large shifts in the biomass is $\gamma_2$, that determines how large a fraction of the background mortality is remineralized directly into N and DOC instead of becoming POM. Increasing $\gamma_2$ increases the amount of dissolved nutrients and carbon in the system which increases the osmotrophic efficiency for picoplankton. However, this value of $\gamma_2$ is highly uncertain, and cell mortality is treated quite simply here because of limited mechanistic understanding (Andersen and Visser, 2023). Apart from cF, $\gamma_2$ is the only other parameter where values are significantly

680 different between the two sites. Values for $\gamma_2$ are larger at Station ALOHA than at CCE, indicating that a faster remineralization of organic matter is required at Station ALOHA. It is clear from the global sensitivity study that developing a clear mechanistic understanding of the fate of cell mortality should be an important priority. Fortunately, a mechanistic model for organic matter accumulation has recently been developed which may be a way to improve the NUM model accuracy in future versions (Zakem et al., 2021).

685  Apart from the parameters described above, the model includes better established parameters that result in a relatively large sensitivity while also influencing the entire size spectrum. Of these, $\sigma,\beta$ defines the shape of the prey-predator size distribution, and $\alpha_{max}$, $\alpha_R$ controls the biosynthesis. In contrast, the effect of $\varepsilon_L$ (light uptake efficiency) mainly influences picoplankton's affinity for photosynthesis.

  Despite the model sensitivity to parameter changes, non-linearity and system bifurcation, the model appears to be relatively

690 stable within the optimized restricted parameter spans identified based on comparison with CEE observations. Within the restricted spans, no parameter combinations seem to perform significantly better than others for the chosen metrics. We recognize however that further local parameter sensitivity investigation can be useful with the current knowledge about the most important parameters gained from the global sensitivity study.

  An underlying premise in our validation is that we compared the model results of a water column setup with annual-mean

695 observations averaging nearly 700 km by 400 km including shelf and open ocean. This means that any parameter combination that performs well compared to the mean dataset will surely be less than optimal at some of the individual stations or at specific times of the year. Ongoing work is evaluating the NUM model in a regional ocean model where smaller variations along shelf and especially across the shelf can be resolved, and in settings were data permits resolution of seasonal variability.

700 **5.3 Areas of improvement**

  We note that the optimal parameters spans have been determined with a water column model without vertical advection. CEE is particularly influenced by upwelling advection while Station ALOHA is influenced by convergence and downwelling advection. This difference is likely a significant factor contributing to the model deficiencies at ALOHA. Indeed the 100,000 random simulations at CEE tend to produce too low nitrate concentrations between 50 m and 200 m

705 depth. This indicates that the model is missing additional upwelling that could push the nutricline up. It may be reasoned, that if there had been more physical upwelling in the model, the higher nutrient loading and presumably growth would mean that a new set of optimal parameter combinations would need to result in less biomass production to fit the observed

biomass. The implication of more efficient biomass downregulation by perhaps more export would mean that for ALOHA, there would be more export driving the system more oligotrophic further enhancing the picoplanktonic biomass and lessen nano and microplankton. In fact, we see in the nutrient profiles that ALOHA has too high nutrient levels from 100 m and deeper. More downwelling advection in the model setup for ALOHA would push the nutricline down and result in a more oligotrophic system, perhaps shifting the ecosystem toward more picoplankton. Regardless, future investigating including a full two-way cross validation should explore NUM in a 3D circulation mode to alleviate model physics deficiencies of the current water column setup.

In the NUM model, there is only one generalist functional group where small to large are defined by the same parameter combination. This means that the smallest sizes, that are essentially bacteria in size, are modeled with the same set of parameters as larger eukaryotic phototrophs. It is well known that there is a myriad of different species of bacteria optimized with different metabolic strategies, optimized with different cell membranes, and with no mitochondria. For example, while the Prokaryotes *Synechococcus* and *Prochlorococcus* are of similar size the former inhabit the surface waters at Station ALOHA while the latter live at low light conditions near the nutricline (Wu et al., 2022). Further, while having quite different modes of life, their resource uptake and growth is also significantly different from for example pico- or nano-eukaryotes. In fact, large meta data analyses show very different allometric scaling of metabolic rate as function of body mass (size) (Delong et al., 2010). Prokaryotes show superlinear scaling with a power of 1.7, while eukaryote protists have linier scaling with a power of 1. Thus, empirical observation seem to suggest that the parameters regulating biosynthesis in NUM may need to respond more strongly to size in the picoplankton end of the spectrum (cf.Delong et al., 2010). In fact, our global sensitivity study revealed that the parameter regulating biosynthesis ($\alpha_{max}$) is among the most important parameters (Fig. 8). We furthermore found that the model in general could not capture picoplankton biomass in the oligotrophic system. However, the best fit between model and observations is with low $r^*_D$ which increases the efficiency of the picoplanktonic community. If biosynthesis in the picoplankton range is modeled as more efficient than in for larger sizes, it potential would upregulate the microbial loop and result in more picoplankton biomass.

Another aspect related to too little pico-biomass under oligotrophic conditions may be related to the model treatment of DOM. Currently the model use DOC contributing only to osmotrophic heterotrophy. However, labile DOM has a DOC:DON ~5-15 (Zakem and Levine, 2019). This means that under oligotrophic conditions the model osmotrophic bacteria are potentially nutrient limited missing an important source of nutrients that could boost the pico-microbial loop thereby increasing $B_{T-pico}$. Adding an explicit or implicit treatment of labile DON would likely result in better performance (cf. Zakem and Levine, 2019). Other recent studies have shown that the picoplankton *Prochlorococcus*, while predominantly phototrophic, is capable of osmotrophic mixotrophy at low light conditions, and that labile DOM additions under low light boots the growth significantly (Wu et al., 2022). The experiments reveal that significant *Prochlorococcus* growth and biomass in the deep chlorophyll maximum is likely sustained by both light and DOM. Such additive substrate would increase the model picoplankton growth rate and boost $B_{A-pico}$ to better match observations.

The simplicity of the NUM model puts some limitations on its use in some environments. The model does not yet include oxygen nor reduction-oxidation reactions as in some trait-based models (cf. Zakem et al., 2020b; Zakem et al., 2020a). This has implications for the large phagotrophs or higher trophic level that are therefore not restricted in their respiration if for example oxygen is low. Using the model below the photic zone in upwelling systems and for investigating low-oxygen environments would require implementation of oxygen, a development that is underway. The model ecosystem is currently not limited by other nutrients than nitrate such as iron or phosphate (cf. Serra-Pompei et al., 2022; Serra-Pompei et al., 2020). It might also be possible to capture more details of the ecosystem by parameterizing or adding additional functional groups such as diatom and bacteria, but these refinements come with a computational cost. Overall, the NUM model is fast and has the benefit of being able to resolve mixotrophy in organisms and shared predation, aspects attracting increasing attention in trait modelling (Wu et al., 2022; Casey et al., 2022; Follett et al., 2022). Our analysis shows that the model – overall and despite its simplicity – is remarkably stable within a wide range of parameters, and usable for a user without intimate knowledge of the parameter settings.

## 6 Conclusion

We have validated the generalist unicellular component of the NUM ecosystem model framework in a water column setup for two sites - a high productivity upwelling system and an oligotrophic downwelling system. With optimization of the range of 23 free parameters, the unicellular component of NUM, despite its simplicity, can capture the size distribution of the planktonic ecosystem and its nutrient profile over a broad range of parameter values. The model reasonably reproduces the nutrient profile despite its simple POM and degradation formulation. For the California Current system (CCE) we find seven optimal parameters combinations that are quite different but almost perfectly match the distribution of each of the size groups as function of increasing $AC_{bio}$. Validation against ALOHA overall indicate that by restricting the parameters based on the optimal parameters for CCE and increasing the microbial loop (increasing $\gamma_2$) and focusing on predation, there is a reasonable match to the overall trends in biomass for the different size classes and the nutrient profile. We find there is a tendency for NUM to underestimate pico- and nanoplankton biomass at both sites, indicating that osmotrophy, nutrient uptake and/or mixotrophy in the lower range of the picoplankton group require further development.

Despite its simplicity, the NUM framework is remarkably stable within the identified restricted parameter ranges and likely well suited for modeling poorly known regions and evolutionary scenarios where first-principles trumps details.

## 7 Code availability

The NUM model used in this analysis along with scripts for running experiments, analyzing results and data, and plotting figures is available at https://github.com/trinefrisbaek/NUM_0.91_ModelEvaluation

(https://zenodo.org/doi/10.5281/zenodo.10844336). The readme file contains a list of relevant scripts for running and plotting files. The original NUM code analyzed in this paper is available at https://github.com/Kenhasteandersen/NUMmodel/releases/tag/v0.91. The simulations are done with the MITgcm_2.8deg transport matrix that has to be downloaded separately from

http://kelvin.earth.ox.ac.uk/spk/Research/TMM/TransportMatrixConfigs/.

## 8 Author contributions

TFH ran the simulations, edited the original NUM model, performed the analysis, and created the figures. CJB and TFH conceptualized the study and processed the observational datasets. TFH prepared the manuscript with contributions from all
co-authors. All authors participated in discussions and provided valuable ideas.

## 9 Competing interests

The authors declare that they have no conflict of interest.

## 10 Acknowledgement

We thank Michael R. Landry for sharing his data from Station Aloha and his insight into the sampling method, as well as
James Wilkinson and the team behind the measurement from the CalCOFI cruises. This work was supported by Villum Fonden Grant 16518, by VKR Center for Excellence "Ocean Life", Independent Research Fund Denmark (10.46540/2032-00265B), and Simon's Foundation grant 931976. CJB secured computation resources on the PALEOK cluster and Computerome 2.0 clusters, at University of Copenhagen.

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
