# Peer review of "The unicellular NUM v.0.91: A trait-based plankton model evaluated in two contrasting biogeographic provinces."

_Geoscientific Model Development, 2024_

## Author Response (AR1)

**Response to Anonymous Referee1**

Thank you for your review and much appreciate the time you have spent. Your comments have greatly improved the paper where we have tried our best to answer thoroughly. Below, the original comments are in black and our response is written in blue.

General comments (overall quality):

The paper is well-written, well-organized, and easy to follow. The methods are clearly explained and many of the questions I had about the model were answered in the Supplement. The analysis has been conducted thoroughly and logically. The results are interesting not just for NUM users but also for those who might be interested in what aspects of a trait-based model might be the most uncertain. It should be published. My only major critique is to suggest that the authors go beyond just the description of the model uncertainty and parameter sensitivity and add some discussion of why the model might be more sensitive to some parameters than others, and what, for example, those three model parameters without overlapping optimized space might mean. Does this suggest areas in which model structure itself might be uncertain? Section 5.1 ("areas of improvement") basically lists areas in which the model could be made to include more processes or modify its descriptions, and in some places this discussion is linked to the sensitivities uncovered by the analysis, but this could be extended. For example, one of the parameters that is significantly different between the two optimizations is gamma_2 (discussed on line 599), which controls DOC and N supply, and so it would be interesting to develop a connection between the uncertainty of this parameter and the discussion of how DOC in the model may be described differently on lines 643-653. In short, the paper could use a bit more high-level synthesis so that it can be more useful in assessing the certain vs. uncertain processes (rather than simply parameters) in trait-based modeling more generally. If this really aims to be a universal, first-principles model, what needs to be changed so that the optimized parameter space is the same for both environments? What aspects are not yet universal?

Thank you for your general comments. Per your suggestions we have added a more high-level synthesis of the parameters that hast the ability to make the largest shift in the model result. This links to a new qualitative parameter process uncertainty estimate that has been added in table 1. We have moreover been more specific in our DOC and N formulation to eliminate any confusion. The comments posted here are incorporated in the specific comments below, so we have answered them there.

Specific comments (individual scientific questions/issues):

L. 18: "Simplest" was confusing to me. Perhaps "in their idealized form" rather than "simplest"? Since below you say that many parameters are not constrained, to me that

means that these models aren't yet exactly defined only by first principles (which by definition are not uncertain).

Yes, thank you. It is now corrected

L. 19: "physics, geometry, and evolution." Should you add metabolism/chemistry here? You mention chemical limitations in the next line.

Yes. "Chemistry" is added to the list.

L. 25-27: I would be more specific about the change in oceanographic setting, and perhaps you could be less specific about the SCC, and rather call them (as you do in the conclusions) a more productive upwelling vs. a more oligotrophic downwelling system.

Thank you for the suggestion. We have clarified the sentences:

"...can capture the general features of the pico-, nano-, and micro planktonic ecosystem in a high-productivity upwelling system. [...] Finally, the model responds correctly in an oligotrophic downwelling system using parameters fitted to the upwelling system."

L. 59: "Hereafter we address the parameters" was confusing to me. Could you make it more clear here that you first evaluate the model's ability using default parameters, and then second, variation in the parameters?

Yes, we have changed the formulation to be more specific:

"Specifically, we start by evaluating the model's ability to capture the size structure of the planktonic biomass at the California Current Ecosystem (CCE) (California-Current-Ecosystem-Lter and Landry, 2019; Taylor and Landry, 2018), using default model parameters. Hereafter we evaluate how the parameter uncertainty effects the model sensitivity."

Table 1:

--A default assimilation efficiency of 0.8 is high for heterotrophic metabolism that uses organic C for energy -- is that different than what is being presented here?

Yes, you are right. Thank you for pointing this out. We are using the default values that are used in Andersen and Visser (2023) where it is the gross assimilation efficiency. In the future, this value should be reduced as also reflected in the optimization done here. Here, we are testing assimilation efficiencies from 0.1 to 0.9 and we find that the good solutions are with a value below 0.4.

--solubilization length scale is blank for the default parameters -- ?

Yes, thank you. We have included a reference value and a reference

--Light attenuation by POM also blank for default and same value for min and max -- ?

--POM sinking coefficient -- seems the max value is wrong.

POM was not included in the previous version of the NUM model where the reference values are from. The values for the three parameters concerning POM were misleading. Thank you for noticing. We have now changed the parameters and added a footnote:

"[3]POM was not included in previous versions of the NUM model and the parameters written in the reference value signify the values used in the initial evaluation of the model. Based on arguments in supplement S1, a $k_{POM}$ value of $3 \times 10^{-5}\,m^2\,mg\,C^{-1}$ is used for all simulations in this article. The choice in POM sinking coefficient and exponent result in a sinking speed of 0.01-3 m day$^{-1}$ for the smallest POM size classes and 1-200 m day$^{-1}$ for the largest, using the formulation for POM sinking in supplement S1."

--I don't know what m+/m- means.

A footnote has been added to table 2 where it is used

"[1]$m^+$ and $m^-$ is mass of the upper and lower limit of the size bin"

Model Description (section 2): If the DOC and POM module is new here, I think this could be better emphasized in the description of the model. Describe this first (rather than last in 2.2.4), and in more detail, since the remainder is a summary from earlier publications (Serra-Pompei etc.). I was very confused about the discussion of "nutrients and DOC" in the sections above (starting on line 106).

General: DOC and N: I am a bit confused by why DOC and N are produced -- are you implicitly assuming that DON is rapidly converted into inorganic form? Why is POM assumed to have N content (I assume this is why you call it POM instad of POC) while DOC isn't? Why not call it DOM? Are you representing heterotrophic osmotrophy, using DOC as the energy source? I could use a clearer description of this module overall, what microbially driven processes are explicitly vs. implicitly resolved.

Thank you for your comment. We agree that there should be a better description of the new modules. In the beginning of section 2 we have added the following sentence:

"Section 2.2 describes the unicellular module and parameters while section 2.3 describes the new DOC and POM modules, and the parameters used therein. "

And we have changed section 2.2.4 to its own section 2.3 and have extended the description to explain the different dissolved and particulate compartments:

**2.3 Dissolved organic carbon and particulate organic matter**

This version of the NUM incorporates both dissolved and particulate matter in a simplified approach (Fig. 1). Dissolved nutrients, both inorganic and organic N containing, are modelled as one dissolved N pool, while dissolved organic carbon (DOC) is modelled separately. The particulate matter (POM) contains both C and N in a fixed ratio. Dead cells, feeding losses, and higher trophic level mortality produce both particulate organic matter (POM) and dissolved constituents (DOC and N). Note, that the choice of pooling inorganic and organic N in a single pool means that the microbial consumption/remineralization of N is not explicitly resolved as dependent on osmotrophy. In contrast, consumption of DOC as an energy source for heterotrophic osmotrophy is explicitly modelled as presented above (section 2.2.1). The pool of DOC in this model represents "labile" DOC. The division between the particular and dissolved fractions are determined by the γ parameters ($\gamma_2$, $\gamma_F$ and $\gamma_{HTL}$), which describe how much of each flux (mortality, feeding losses, and higher trophic level mortality) are routed to the dissolved fractions, with the remaining losses transferred to POM. Particulate organic matter is here divided into two different size fractions (a number that can readily be increased in future applications). POM derived from dead cells and feeding losses is transferred to the largest POM size fraction, which is smaller than the size of the original cell. POM from higher trophic level mortality is transferred into the largest POM size fraction. POM sinks with a size-dependent velocity, described as a power function with the parameters $v_1$ and exponent $v_2$ (Eq. (9)). POM is assumed to remineralize directly to the dissolved N and DOC pools. This process of remineralization is not explicitly microbial cell related in the model but occurs at a constant rate determined by the inverse of the solubilization length scale (**a**) as $\text{rem}_{POM} = a w_{POM}$. The model formulation of nutrient, along with DOC and POM modules are given in Supplement S1."

Table 2 and general: I am guessing that "m" is the mass of each size group, but I couldn't find this stated anywhere in the main text.

Yes, it is now added to the text. Thank you

L. 179: Again mention that you are doing the initial evaluation of just the default parameters.

Yes, thank you. We have included it in the sentence:

"We initially perform a general validation of the model with default parameters against the mean biomass size spectrum and nutrient profile for the two locations"

L. 193 and more broadly: I'm not quite sure why we would be surprised that varying the parameters among only the more restricted parameter set would do anything but improve model performance.

The question is: does it take a very specific combination of parameters to obtain good model fit or will any parameter combination, if they are within this restricted parameter span be resulting in a good fit? One could imagine that the non-linearity of the model results in just a few "lucky shots" while parameter combinations slightly different would result in poor model fit. The line is now modified to clarify this point:

"This subanalysis allow us to determine if only very specific combination of parameter results in good model fit or if model performance is increased by simply reducing the parameter span."

L. 195 and more generally: Why only 7? What criteria were used? Apologies if I missed this somewhere.

We have removed the reference to the number 7 here as it is confusing. We have also added a description of the criteria that was used in the end of section 3.2:

"Ideally, the optimal successful model simultaneously has $COR_{m-o} > COR_{iao}$ and $RMSd_{m-o} < RMSd_{iao}$ for all biomass size categories. As is clear below, no model results fulfill both criteria for all biomass size categories. Instead, we isolate the model results that fits these criteria for at least 8 out of 10 size categories and has biomass in ACbio-bin up to at least 40 $\mu gCl^{-1}$ for CCE and 15 $\mu gCl^{-1}$ for HOT. For the solutions that fulfill these criteria we sort them according to their $COR_{m-o}$ and $RMSd_{m-o}$ and make a visual qualitative assessment in comparison with observations (cf. Fig 2)."

L. 313: "no model results fulfill both criteria for all biomass size categories." I really want to know why this is! Could you comment in discussion what this might mean? With your intimate knowledge of the model, could you comment on what part of the model might not yet be universal?

Thank you for this comment. We have included discussion on the different parameters that create the largest model uncertainties (See response for comment on line 598-601). We hope this makes the model's insufficiencies clear.

L. 327: But the majority of the 60 uM DOC is recalcitrant. How would the model solutions change if initialized DOC were at 1 uM? How much DOC is remaining in the steady-state model solutions?

We have initialized the model with different DOC values, but the initial value has no impact on the steady-state model result. Our steady-state solution shows annual fluctuations in the surface between 0.5 and 1.8 $\mu m$ with an average DOC value of approximately 1 $\mu m$. We have not focused on DOC in this paper as the handling of DOC is crude, because of the lack of mechanistic understanding. However, we have noted this in the discussion and added a reference to Zakem et al. (2021) as a possibility for future implementation. Moreover, we have modified the sentence for clarity:

"DOC is initialized with a value of 60 µmol kg$^{-1}$ (Zakem and Levine, 2019; Sarmiento and Gruber, 2006; Letscher and Moore, 2015). DOC rapidly decreases to dynamic steady state with an annual mean value of $\sim 1 \pm 0.5$ µmol kg$^{-1}$."

And we have added a sentence in model stating that modelled DOC is representing labile DOC.

L. 383/Fig. 4: Why might the model produce these distinct groups? Could it be something to do with predator-prey oscillations? Can you comment on what part of the model structure/equations might lead to this behavior?

This is an excellent and intriguing question. It is most likely combinations of parameters that may produce different quantized biomass states. We have added a sentence:

"Interestingly, the result of the simulations falls within three distinct groups for $B_{A-pico}$, where some parameter combinations produce a much better correlation with observations than others. That $B_{A-pico}$ fall in three groups may be related to the biomass quantization also found in observations and other size-structured planktonic ecosystem models (Moscoso et al., 2022; Schartau et al., 2010)."

However, we defer an in-depth analysis of this phenomenon as it will take a very large effort that we deem is beyond the scope of this article.

Fig. 6 panel c: It is neat that the seven solutions capture this peak behavior -- is there any insight into what in the model might produce this? Again, getting at a bigger synthesis of these insights.

We thank you for this suggestion. It would be very interesting to get insight into how parameter combines to create this behavior. We have performed a cluster analysis to investigate the parameter relation but due to high level of non-linearity in the model and possible size quantization, the suggested analysis will have to be included in a more detailed study of parameter optimization, based on the findings of our current paper.

L. 495: "parameter tuning may not be necessary." This doesn't quite make sense to me, because it seems that indeed you have tuned the parameters already by selecting the 7 best models. It's not surprising then that sampling within this more restricted space produces better results.

We have expanded on this point for added clarification:

"Overall, the model results in Fig. 9 demonstrate a notable improvement in model performance for the identified parameter spans in comparison to the full parameter space. While this improvement may seem intuitive it is not necessarily a priori given, considering the models parameters non-linearity. The local sensitivity analysis showed that, even within this restricted parameter space, the impact of varying a parameter is highly dependent on the other parameters (Fig. S2). The restricted parameter space

could therefore, in theory, have resulted in the same degree of model misfits as the full parameter span with only a few acceptable solutions generated by very specific parameter combinations. That the model performance is enhanced by restricting the parameter span suggests that further detailed parameter tuning may not be necessary to achieve reliable results from the NUM model"

L. 524 (and more general topic of remineralization): What does it mean that some of the dead matter is "directly remineralized back to nutrients"? Are you referring to DOC as a nutrient here? What process is causing this remineralization? Are these microbial types that are not explicitly resolved? How would this affect biomass distribution? Are you assuming just that the POM consumers are implicit? This links back to setting up the reader with a clearer description of the new DOC and POM modules.

This comment links back to the comment on Model Description earlier. We have added a new section (section 2.3) that hopefully makes this sentence in the discussion clear now.

L. 571: "only 7 optimal" -- again, how are these 7 determined? Why are there 7? What was the threshold?

See comment for L195 above.

L. 598-601: You are in the discussion now, so rather than just again repeating the same results, can you discuss or even speculate about what it means that these are the most uncertain parameters. Are these in line with what we think are the most uncertain processes? What insights have been revealed? Does it say anything about what the field should be studying or observing more closely?

Thank you for this comment. We have extended the parameter discussion with a discussion of the most uncertain of the parameters that have been reviled to be important through the global sensitivity analysis:

[revised manuscript text omitted]

Technical corrections:

L. 25: "in the" rather than "at the" SCC.

Yes, thank you, it is corrected.

L. 28: I woul not start another second paragraph within the abstract, but perhaps this is a typo. Also, perhaps just change "accessible for the general non-expert" to "broadly accessible."

Yes, thank you, it is corrected.

L. 181: I would start the new paragraph at "The investigation.."

Yes, thank you, it is corrected.

L. 194: I don't know what "with outset" means (here and the next line).

Outset is here meant as "starting point". We cannot find a word that fits better and have instead tried to clarify that the analysis has its "outset" in a specific initial parameter combination:

"The second subanalysis is a set of local sensitivity analyses where the model's sensitivity toward each of the parameters is evaluated separately with outset in an initial parameter combination (Zhou and Lin, 2008). The local sensitivity analysis is made with outset in the initial parameter combinations that performs best for CCE"

Fig. 2: At first glance it seemed as if the x and y axes were exactly the same for plots a and b. Could the y axis be changed to something like "size class biomass"?

Yes. We have changed the y-axis to "Size-separated autotrophic biomass $B_A$" and the y-axis for subplot e and f to "Size-separated total biomass $B_T$". The same change has been done to all other figures with the same axis and these have been updated

L. 277: Should this be B_A-micro?

Yes, Thank you. That was an important correction.

L. 301: Instead of "identify", do you mean "define"? Also, "are" instead of "is" for model results.

Yes, thank you, it is corrected.

L. 303: take out "its" from "its STDs" and just write out "standard deviation"

Done, thanks.

L. 380: grammar problem: "size groups all overestimate" does not make sense.

The sentence is now changed to "The amplitude of variation in the size spectrum is overestimated for all size groups of pico- and nanoplankton"

L. 395: "lack of model results with.." doesn't make sense -- something like "model does not capture the biomass concentrations at …"

We have substituted this text with:

"Generally, the model does not capture the occurrences of high $AC_{bio}$ concentrations ( $AC_{bio}$ above approximately 100 µg C/l)"

L. 523: here you mention "partly aF" and discuss, and then in other sections of the manuscript you just discuss gamma_2 and cF without aF. This led to my incorrect summary above (that I am just now realizing) that there are only two parameters that do not overlap. Just noting this. aF does overlap somewhat, and some other parameters (like r_D) also don't overlap much, so aF not qualitatively different than others in this light. I realize this is why you end up just discussing two. Perhaps make this clear here: "aF is one example of where there is some, but little, overlap"...

Thank you, it was confusing. We have now changed the formulation to:

"Interestingly, most of the parameters for these four simulations fully overlap with parameter for the best solutions at CCE (Fig. 7). While some parameters such as aF and $\alpha_{max}$ only partly overlap, the only parameters that significantly differs between the two sites are $\gamma_2$, cF that both have higher value at Station ALOHA than CCE"

**Response to Anonymous Referee2**

Thank you for your review and much appreciate the time you have spent. Your comments have greatly improved the paper where we have tried our best to answer thoroughly. Below, the original comments are in black and our response is written in blue.

Overall: The manuscript presents a thorough overview of a newly developed generalized trait-based Nutrient-Unicellular- Multicellular (NUM) model of the size distribution of plankton, which accounts for a mixotrophy, i.e. both autotrophic and phagotrophic (heterotrophic) growth. The manuscript includes an evaluation of model performance based on 100,000 simulations with varied parameter values for each of two contrasting oceanic sites: 1) oligotrophic station ALOHA, with persistently low nutrients and biomass and overall smaller size distribution of plankton 2) coastal/shelf CCE, with relatively higher nutrients and biomass and accordingly relatively more larger plankton. Based on the best results so obtained, the range of parameter values was narrowed and further optimized (by randomly sampling the parameter space), and this was shown to yield further improvements in the agreement between model results and the observed size distribution of plankton. A one-way cross validation, using the parameter values obtained for CCE to simulate ALOHA, revealed reasonable agreement between model results and observations. However, the reverse cross-validation was not presented. The combination of a size-based model, formulated to be comparable to typical oceanic observations, with a representation of mixotrophy make this a promising model framework for exploring hyoptheses about observed patterns.

As the authors note in the Discussion, one important feature of the trait-based approach is its potential for general applicability, even to sites or scenarios where limited (or no) observations are available. To some degree, the cross validation presented supports this general applicability of the new NUM model. That case could be made stronger if the reverse cross validation (using parameters tuned to station ALOHA to simulate CCE) were presented.

We thank you for making this very valid point on cross-validation. This topic is also raised in the specific comments below and we therefore address the comment below.

The formulation presented does seem to provide a generally applicable modeling framework, which could be applicable to a wide variety of oceanic sites/regions. The parameter sensitivities identified by the authors, and some insightful information about the how the model could potentially be modified to better capture the physiological response of prokaryotic phytoplankton, provide a useful basis for future studies to enhance the applicability of the model framework.

Specific Comments:

Line: 53: "how much tuning the model acquire...". Should that be "require", or perhaps better yet "how much the results depend on the specific values chosen for parameters" ?

Yes, thank you. We have changed "acquire" to "require", keeping the sentence short.

Line 192-193: "This subanalysis allow us to determine how much model performance is increased by simply reducing the parameter span."  Does this refer to the fact that randomly smapling parameter values over a narrower, rather than a wider, range will most likely find parameter sets that better capture the observations?  In any case, revise for to clarify the meaning.

These lines refer to a central question: does it take a very specific combination of parameters to obtain good model fit or will any parameter combination, as long as they are within this restricted parameter span be resulting in a good fit? The sentence has been modified to clear up any confusion:

"This subanalysis allow us to determine if only very specific combination of parameter results in good model fit or if model performance is increased by simply reducing the parameter span."

Line 277: "autotrophic microplankton ($B\_T$-micro)."  Shouldn't the subscript be "A": i.e., "$B\_A$-micro")?

Yes, thank you for noticing. It has been changed.

Also, a Table summarizing the notation for the various classes of biomass considered, as well as the different metrics (correlation and RMSE errors) would be helpful. It is a little complicated and not obvious what these differnet subscripts mean. I had to go back and search to refresh my memory while reading the manuscript.

Good idea. We have added two new tables to include these metrics (Table 3 and 4)

Line 521: "Figure 11 show a set of simulations that performs particularly well for Station ALOHA. " Were those parameter sets (and corresponding simulations) obtained from the narrowed search based on the parameter range identified from the intial simualtions of CCE?

No, they were from the first-level 100,000 simulations with conditions from Station ALOHA. This has now been detailed in the text:

"Figure 11 shows a set of simulations from the first-level random parameter study that performs particularly well for Station ALOHA"

Finally, although the authors have already clearly gone to an extensive effort, would it be possible to include a brief presentation of the cross validation using parameters tuned for station ALOHA (as the author have already identified) to simulate CCE? The would provide additional information about the portability of the model formulation. It could also be put into the supplemental and only briefly summarized in the main text, to avoid making the manuscript too long.

Thank you for this valid point. While we agree that it is an important analysis of the model performance, we suggest that it would be more relevant to do this cross validation in a later 3D setup of the model. In the discussion we describe how lack of advection in the water-column setup puts some limitations on how well the fit can be (section 5.3). We have added the following sentence to the discussion to comment on this point:

"...future investigating including a full two-way cross validation should explore NUM in a 3D circulation mode to alleviate model physics deficiencies of the current water column setup. "

Minor: the manuscript includes numerous mis-spellings and minor grammatical errors, which can be corrected later, during production.

Thank you. We have spell-checked the document once more.